# DYNAMIC SPARSE TRAINING
# WITH STRUCTURED SPARSITY

**Mike Lasby**[1], **Anna Golubeva**[2,3], **Utku Evci**[4], **Mihai Nica**[5,6], **Yani A. Ioannou**[1]
[1]University of Calgary, [2]Massachusetts Institute of Technology, [3]IAIFI
[4]Google DeepMind, [5]University of Guelph, [6]Vector Institute for AI [*]

## ABSTRACT

Dynamic Sparse Training (DST) methods achieve state-of-the-art results in sparse neural network training, matching the generalization of dense models while enabling sparse training and inference. Although the resulting models are highly sparse and theoretically less computationally expensive, achieving speedups with unstructured sparsity on real-world hardware is challenging. In this work, we propose a sparse-to-sparse DST method, Structured RigL (SRigL), to learn a variant of fine-grained *structured* N:M sparsity by imposing a *constant fan-in* constraint. Using our empirical analysis of existing DST methods at high sparsity, we additionally employ a neuron ablation method which enables SRigL to achieve state-of-the-art sparse-to-sparse structured DST performance on a variety of Neural Network (NN) architectures. Using a 90% sparse linear layer, we demonstrate a real-world acceleration of $3.4\times/2.5\times$ on CPU for *online inference* and $1.7\times/13.0\times$ on GPU for inference with a batch size of 256 when compared to equivalent dense/unstructured (CSR) sparse layers, respectively.

## 1 INTRODUCTION

Dynamic Sparse Training (DST) methods such as RigL (Evci et al., 2021) are the state-of-the-art in sparse training methods for Deep Neural Networks (DNNs). DST methods typically learn *unstructured* masks resulting in 85–95% fewer weights than dense models, while maintaining dense-like generalization and typically outperforming masks found via pruning. Furthermore, sparse-to-sparse DST algorithms are capable of employing sparsity *both during training and inference*, unlike pruning and dense-to-sparse DST methods such as SR-STE (Zhou et al., 2021) which only exploit sparsity at inference time.

While models trained with DST methods are highly sparse and enable a large reduction in Floating Point Operations (FLOPs) in theory, realizing these speedups on hardware is challenging when the sparsity pattern is unstructured. Even considering recent advances in accelerating unstructured Sparse Neural Networks (SNNs) (Gale et al., 2020; Elsen et al., 2020; Ji & Chen, 2022), structured sparsity realizes much stronger acceleration on real-world hardware. On the other hand, structured sparse pruning often removes salient weights, resulting in worse generalization than comparable unstructured SNNs for the same sparsity level (Fig. 1a). Our work presents a best-of-both-worlds approach: we exploit the DST framework to learn *both* a highly-sparse *and* structured representation while maintaining generalization performance. In summary, our work makes the following contributions:

1. We propose a novel sparse-to-sparse DST method, Structured RigL (SRigL), based on RigL (Evci et al., 2021). SRigL learns a SNN with constant fan-in fine-grained structured sparsity (Fig. 1a) while maintaining generalization comparable with RigL up to a high sparsity level (99%) for a variety of network architectures. This structure is a particular case of "N:M sparsity" which requires $N$ out of $M$ consecutive weights to be non-zero (Mishra et al., 2021).
2. Our empirical analysis shows RigL, at sparsity levels $> 90\%$, ablates whole neurons. By allowing neuron ablation in SRigL, we match RigL generalization even in this high-sparsity regime.
3. We enable neuron ablation in SRigL across all sparsity regimes. We find this structured sparsity is complementary to the constant fan-in sparsity in improving real-world inference timings while maintaining generalization comparable to unstructured DST methods.

---
[*]{mklasby,yani.ioannou}@ucalgary.ca, golubeva@mit.edu, evcu@google.com, nicam@uoguelph.ca
Our source code is available here.

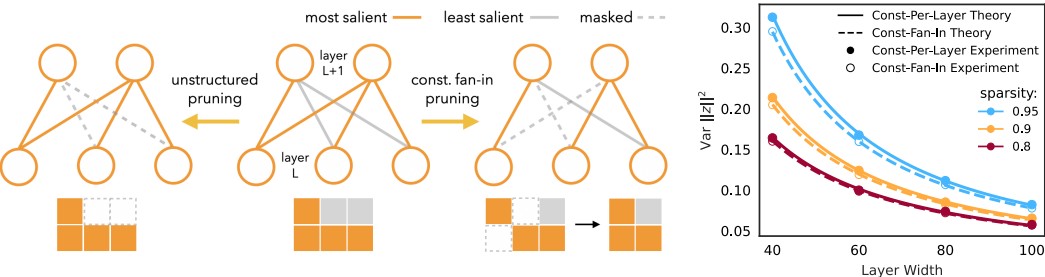

**(a)** Constant fan-in pruning v.s. unstructured pruning.    **(b)** Output-norm variance analysis.

**Figure 1:** (a) **Constant fan-in** pruning keeps the most salient weights *per neuron*, while unstructured pruning keeps the most salient weights *per layer*. A constant fan-in weight matrix has the same number of non-zero elements (here 2) per column allowing condensed representation. While pruning may remove salient weights affecting generalization, with SRigL structure and weights are learned concurrently. (b) **Output-norm variance**: Theoretical predictions and simulation results (see Appendix A) demonstrating that sparse layers with constant fan-in have consistently smaller output-norm variance than layers with the same sparsity but w/o the constant fan-in constraint.

4. We demonstrate that constant fan-in sparsity enables a compact representation that is not only parameter- and memory-efficient, but also amenable to real-world acceleration. We observe significantly reduced real-world timings for online inference using our CPU-based PyTorch implementation and for batched inference using a GPU-based implementation from Schultheis & Babbar (2023) over dense and unstructured baselines.

## 2  RELATED WORK

**Dynamic sparse training**    Unlike with pruning, where weights are typically pruned after the dense network was trained (Han et al., 2015; 2016), or at initialization (Wang et al., 2020), DST methods learn the sparse connectivity during training by periodically adding and removing weights based on various saliency criteria. For instance, Sparse Evolutionary Training (SET) (Mocanu et al., 2018) removes weights with the smallest magnitude and adds weights randomly; similarly, RigL (Evci et al., 2021) prunes weights with the smallest magnitude and regrows weights that have large-magnitude gradients. Liu et al. (2021c) further improved the original RigL results by increasing the extent of the parameter space explored by modifying the sparse connectivity update schedule and drop rate.

Many recent works have examined the effect of different grow and prune saliency criteria on unstructured DST approaches, including SET, Deep Rewiring (DeepR) (Bellec et al., 2018), Sparse Networks from Scratch (SNFS) (Dettmers & Zettlemoyer, 2019), Dynamic Sparse Reparameterization (DSR) (Mostafa & Wang, 2019), Top-K Always Sparse Training (Top-KAST) (Jayakumar et al., 2020), and Memory-Economic Sparse Training (MEST) (Yuan et al., 2021a). In Section 4, we compare SRigL to several of these methods. While the above-noted DST methods are highly effective at finding SNNs which reduce theoretical inference cost, they result in unstructured SNNs which are difficult to accelerate in practice on common hardware architectures.

In a contemporaneous work, Yin et al. (2023) also identified the existence of *sparse amenable channels* in existing unstructured DST algorithms. Their method, Chase, achieves state-of-the-art generalization performance by including a soft memory bound similar to Yuan et al. (2021b) and calculating the saliency of parameters based on global instead of layer-wise statistics. Chase requires that the structured sparsity level be set prior to training. In contrast, SRigL dynamically learns to ablate channels based on the number of remaining weights that are considered salient.

**Accelerating unstructured sparse neural networks**    Elsen et al. (2020) proposed a method for accelerating unstructured SNNs based on one-dimensional tiling of non-zero elements, which demonstrated significant speedups on both Central Processing Unit (CPU) (Elsen et al., 2020) and Graphics Processing Unit (GPU) (Gale et al., 2020). However, like most approaches to accelerating unstructured SNNs, this method relies on imposing structure on an existing sparse weight matrix *after training*. Our method can be considered a way of adding structure to SNNs *during training*, allowing the model to maximally utilize non-zero weights since structure and weights are learned concurrently.

DeepSparse Engine (Neural Magic, 2021) accelerates inference of unstructured sparse networks on CPU by applying several innovations. In Appendix K, we compare our timings with SRigL to the DeepSparse Engine.

**Learning block structured sparsity from scratch**   Block sparsity is a particular type of structured sparsity in which blocks of non-zero weights are grouped together in arrangements that reduce the memory overhead required to store the indices of the non-zero weights. Blocks can be generated out of contiguous weights in 1D (sometimes called tiles) or 2D or by utilizing a fixed number of non-zero weights per row or column group in the case of block-balanced sparsity (Hoefler et al., 2021). Spurred by the success of DST in learning unstructured sparse models, recent works have attempted to apply DST principles to learn block-structured sparsity. Jiang et al. (2022) introduced a novel block-aware DST algorithm known as Dynamic Shuffled Block (DSB). DSB reshuffles non-zero weights into a block sparsity pattern after sparse connectivity updates, thereby improving memory access efficiency. Wall-clock speed-ups of up to $4\times$ were reported with this method; however, generalization performance was reduced compared to RigL at comparable sparsities. Dietrich et al. (2022) applied a modified variant of RigL to BERT models (Devlin et al., 2019). The resulting method is capable of learning models with block-structured sparsity.

**Learning N:M structured sparsity from scratch**   N:M sparsity is a specific form of block-balanced sparsity in which 1D blocks with $M$ contiguous elements contain exactly $N$ non-zero elements. N:M sparsity is particularly amenable to acceleration and several attempts have been made to train models with N:M fine-grained structure using DST methods.

Yang et al. (2022) extended the DST method proposed by Liu et al. (2021b) to train multiple sparse sub-networks sampled from a single dense super-network. Their proposed method, Alternating Sparse Training (AST), switches the network topology between sparse sub-networks after each mini-batch during training. Yang et al. (2022) demonstrated state-of-the-art performance on several typical sparse training benchmarks. However, the dense model weights and gradients are required throughout the majority of training, greatly increasing the overall compute and storage requirements. While AST demonstrated a tantalizing possibility of training multiple sparse sub-networks within a single training loop, the gradual dense-to-sparse training paradigm used by (Liu et al., 2021b) is not directly comparable to RigL or other similar end-to-end sparse DST methods.

Zhou et al. (2021) explored how N:M sparsity can be achieved during training using magnitude-based pruning during the forward pass and a Straight-Through Estimator (STE) (Bengio et al., 2013) on the backward pass. In their method, the dense network weights are projected into a sparse network during each training iteration. The sparse network is obtained by selecting the top-N out of every M contiguous weights and STE is used to propagate the approximated gradients through the projection function. A regularization term is applied to the gradients of pruned weights to reduce instabilities during training. Their approach — Sparse-Refined Straight-Through Estimator (SR-STE) — was applied to networks with N:M ratios of 1:4, 2:4, 2:8, 4:8, 1:16.

Although SR-STE utilizes sparse operations in the forward pass and can find sparse models optimized for inference, it does not reduce the training cost significantly. Specifically, SR-STE training requires (1) storing original parameters in their dense format, and (2) calculating dense gradients during each training iteration. This makes SR-STE training as expensive as the original dense training in terms of memory and compute cost[1]. On the other hand, DST methods such as RigL, and our proposed method SRigL, are capable of end-to-end sparse training and use sparse parameters and gradients throughout training.

**Accelerating fine-grained N:M structured sparsity**   Nvidia (2020); Mishra et al. (2021) introduced the Ampere Tensor Core GPU architecture (e.g. A100 GPUs) and proposed the 2:4 fine-grained structured sparsity scheme that enables SNNs to be accelerated on this hardware *at inference time*. This scheme places a constraint on the allowed sparsity pattern: For every contiguous array of four weights, two are pruned, yielding a 50%-sparse net. The resulting regular structure of the weight matrix allows one to compress it efficiently and to reduce memory storage and bandwidth by operating on the nonzero weights only. Since the focus is on acceleration at inference time, the authors proposed to use the standard method of magnitude-based pruning post training to achieve the 2:4 sparsity. Importantly, this work considered exclusively the 2:4 ratio; other N:M ratios cannot be accelerated on Ampere GPUs.

---

[1]To be precise, SR-STE can use some sparse operations and reduce training cost up to two thirds of the original dense training. However this is still far from fully sparse acceleration for training.

**Constant fan-in N:M structured sparsity**    The constant fan-in constraint represents a special case of N:M sparsity where $N$ is the number of non-zero weights per neuron and $M$ is the dense fan-in for each neuron within a given layer. While commodity hardware acceleration currently exists only for 2:4 sparsity on Nvidia's Ampere and later architectures (Mishra et al., 2021), a constant fan-in constraint can also take advantage of the efficient memory access and throughput increase that N:M sparsity yields, as recently demonstrated by Schultheis & Babbar (2023). Constant fan-in sparsity has several attributes which differentiate it from N:M sparsity:

- Constant fan-in sparsity is more flexible than N:M sparsity, enabling arbitrary global sparsity values to be applied to the mode whereas N:M sparsity is limited to specific sparsity ratios.
- With the constant fan-in constraint, per-layer sparsity distributions such as Erdős-Rényi-Kernel (ERK) can be applied to the model. The ERK distribution has been demonstrated to outperform uniform sparsity distributions by reallocating parameters to layers with fewer parameters (Mocanu et al., 2018; Evci et al., 2021). In contrast, N:M sparsity can only be applied with a uniform sparsity distribution.
- Hardware support for acceleration of N:M sparsity is currently limited to 2:4 sparsity on Nvidia GPUs, offering a modest acceleration on the order of $\times 2$. In contrast, the potential promise of highly sparse models (>=90% sparsity) to be $\times 10$ faster than an equivalent dense model. As we demonstrate in Section 4.4 and Appendix I, our condensed sparse representation with constant fan-in sparsity can achieve significant acceleration over a wide range of sparsities even without specialized hardware.

**Online inference**    In many applications, DNNs are used in an *online* manner, i.e. by using only single inputs and not batches of inputs. Online inference is common in real-time and latency-sensitive applications, or applications without significant numbers of simultaneous requests allowing batching. Online inference, especially for real-time applications, does not typically benefit from accelerators such as GPUs that require host to device transfers, since the cost of the transfer itself often negates any benefit in compute. Accelerating online inference workloads remains an open research problem, with many systems engineering solutions proposed to achieve acceleration (Kumar et al., 2019; Li et al., 2020; Wang et al., 2022; Wu et al., 2020). Our condensed representation CPU implementation, which exploits both structured and constant fan-in sparsity, offers a complimentary, orthogonal solution to these engineered solutions by directly accelerating model inference for single samples.

## 3  METHOD

Our goal in this work is to introduce structural constraints on the sparse mask learned by RigL, in order to make it more amenable to acceleration at inference time while not affecting RigL's generalization performance. We first performed a theoretical analysis to explore the effect of various sparsity distributions with different degrees of structural constraints on the training dynamics of SNNs, detailed in Fig. 1a and Appendix A. Based on this analysis, we did not find any evidence to suggest that the constant fan-in constraint would impair SNN training dynamics and performance, motivating the use of constant fan-in sparsity in our method outlined in Section 3.1.

### 3.1  STRUCTURED RIGL

As motivated by Appendix A, we propose to enforce the constant-fan-in constraint within a sparse-to-sparse DST method to learn structured sparse connectivity from scratch. Specifically, we use RigL by Evci et al. (2021), which can obtain highly sparse networks with generalization performance comparable to their dense baselines.

In brief, the methodology of RigL is to update the SNN connectivity during training by *pruning* weights with the smallest magnitude and *regrowing* those with the largest corresponding gradient magnitude in *each layer*. This occurs in periodic, but relatively infrequent mask update steps throughout most of training. In SRigL, weight saliency must be determined at the *neuron level* (in convolutional layers, at the level of each filter), since we enforce that every neuron (output channel) has the same number of unmasked incoming weights, thereby satisfying the constant fan-in constraint. (Fig. 1a).

However, this approach alone significantly lags behind RigL's generalization at very high sparsities (>90%) and with transformer architectures, as shown in Fig. 3a and Table 4. This is because the constant fan-in constraint has an important side-effect: under a strict constant fan-in constraint, neurons

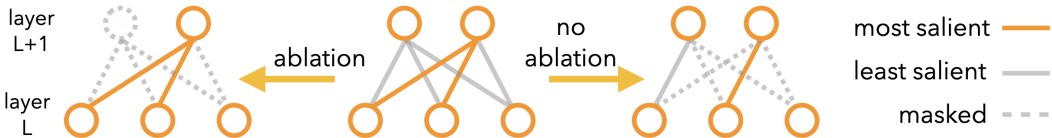

**Figure 2: Neuron ablation.** At sparsity levels over 90%, RigL learns to completely mask (ablate) a large number of neurons within each layer, effectively reducing layer width. Imposing a constant fan-in constraint requires all neurons to have the same number of (non-pruned) incoming weights and therefore inhibits ablation, which results in worse generalization performance than RigL. Allowing SRigL to ablate neurons restores RigL-level performance.

can never be entirely masked (ablated), as illustrated in Fig. 2. At very high sparsity levels this can lead to many neurons that have only 1–2 weights, limiting the capacity to learn complex features and consequently reducing generalization performance. Indeed, at high sparsities we observed empirically that RigL ablates large numbers of neurons (Figs. 3b, 11 and 12). Effectively, *RigL reduces the width of the model at high sparsities to maintain generalization performance*; we believe we are the first to explicitly identify this behaviour within a DST method. To resolve this issue in SRigL, we implement a **neuron ablation method**, allowing SRigL to maintain both a constant fan-in constraint *and* to reduce layer width at high sparsities. We introduce a new hyperparameter, $\gamma_{sal}$, which defines the required minimum percentage of salient weights per neuron. Given a neuron with constant fan-in of $k$, if fewer than $\gamma_{sal} * k$ weights are considered salient by either the drop *or* grow criteria, then the neuron is ablated and its weights redistributed to other neurons within the same layer. Notably this neuron ablation method allows SRigL to exploit neuron ablation structured sparsity *at much lower sparsity levels* than we identified it occurring at in RigL, while maintaining good generalization, as demonstrated in Table 4.

The steps below outline our final SRigL method with neuron ablation. In the following procedure, the first two steps are the same as in RigL, while the other steps are specific to SRigL, containing modifications to include the constant fan-in constraint and dynamic neuron ablation. We first set an ablation threshold $\gamma_{sal}$. Then, for each layer we do the following:

1. Obtain magnitudes of the active weights and gradient magnitudes of the pruned weights; these will serve as prune and growth criteria, respectively.
2. Compute $K$, the number of weights to be grown and pruned in the current step in this layer. We always grow the same number of connections as we prune.
3. Count the number of salient weights per neuron. A weight is considered *salient* if it is in the top-$K$ of *either* the largest-magnitude weights or the largest-magnitude gradients.
4. Ablate neurons that have fewer salient weights than $\gamma_{sal} * k$, where $k$ is the fan-in. Ablation is done by pruning all incoming weights. These pruned weights are redistributed to the remaining neurons in the following steps.
5. Compute the new constant fan-in constraint, $k'$, based on the number of ablated neurons.
6. Prune the $K$ smallest-magnitude weights in the current layer. Note that this pruning criterion considers all weights within a layer rather than pruning only the smallest weights in each neuron.
7. For each active neuron, regrow as many weights as required, proceeding in order of decreasing gradient magnitude, until the target fan-in, $k'$, is achieved.

## 4 RESULTS

We implement SRigL in PyTorch by extending an existing implementation of RigL (McCreary, 2020). We evaluate our method empirically on image classification tasks: on the CIFAR-10 dataset (Krizhevsky, 2009) we train a variant of ResNet-18 (He et al., 2016) suitable for CIFAR-10 and Wide ResNet-22 (Zagoruyko & Komodakis, 2017); on the 2012 ImageNet Large Scale Visual Recognition Challenge (ILSVRC-12) dataset (Russakovsky et al., 2015) — commonly referred to as ImageNet — we train ResNet-50 (He et al., 2016), MobileNet-V3 (Howard et al., 2019), and Vision Transformer (ViT-B/16) (Dosovitskiy et al., 2021). See Appendix C and Appendix D.4 for Wide ResNet-22 and MobileNet-V3 experimental results, respectively.

Unless noted otherwise, we use the same hyperparameter configuration as the original RigL method. A detailed summary of our hyperparameter settings and training details can be found in Appendix D.

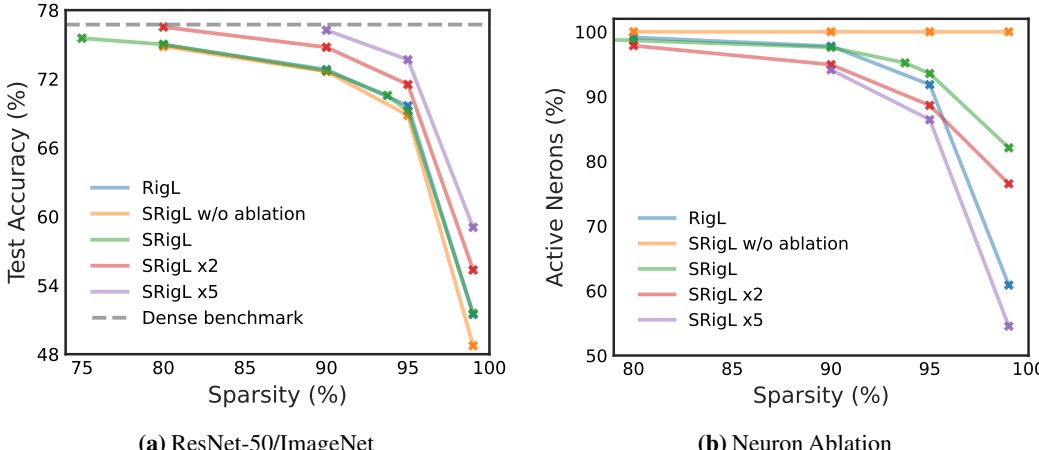

**(a)** ResNet-50/ImageNet  **(b)** Neuron Ablation

**Figure 3: (a) ResNet-50/ImageNet top-1 test accuracy** when trained with SRigL for a range of sparsities is comparable to RigL. Extended training durations of ×2 and ×5 are also reported for SRigL. Results reported are single runs. **(b) Neuron ablation:** The percentage active neurons (i.e., not ablated) following RigL/SRigL training on ResNet-50/ImageNet. RigL ablates a large number of neurons at high sparsities.

We set the ablation threshold, $\gamma_{sal}$, to 30% for all SRigL results, except for our ViT-B/16 experiments. This value was selected based on a hyperparameter sweep performed by training ResNet-18 and Wide ResNet-22 on the CIFAR-10 dataset, see Appendix E.

## 4.1 RESNET-18 TRAINED ON CIFAR-10

We use a variant of ResNet-18 with reduced kernel dimensions and stride in the first two convolutional layers to obtain a model suitable for CIFAR-10; our training regimen generally follows Evci et al. (2021), see Appendix D.1 for more information. We repeat training with five different random seeds for both methods and report the mean and 95% confidence interval compared to a densely-connected benchmark model in Table 2. These results confirm that imposing a constant fan-in constraint during sparse training does not significantly degrade generalization performance of the SNN compared to the RigL method. In Fig. 11 we plot the number of neurons ablated at ablation thresholds of 0%, 30%, and 50% to demonstrate how the $\gamma_{sal}$ hyperparameter can be used to guide the final model width during training.

## 4.2 RESNET-50 TRAINED ON IMAGENET

Our training regimen for the ImageNet dataset generally follows Evci et al. (2021), see Appendix D.2 for more details. We investigate the effect of extended training with ×2 and ×5 the original number of training epochs. We train each model with a single seed and report the results in Fig. 3a and Table 1.

SRigL yields similar generalization performance as RigL across each sparsity and training duration considered. At high sparsities, SRigL with ablation outperforms SRigL without ablation, highlighting the importance of neuron ablation as sparsity increases. Notably, RigL ×5 results at 99% sparsity in Evci et al. (2021) used a dense first layer, unlike all other results reported in Table 1. Despite this difference, SRigL ×5 at 99% sparsity is comparable to the RigL ×5 results. We expect that the 99% sparse models would be improved by using a dense first layer for all SRigL results. Similar to RigL, we observe that SRigL generalization performance improves with increasing training time.

We inspect the connectivity of ResNet models trained with the RigL method and find, as shown in Fig. 3b, that at 95% sparsity 10.9% of neurons are removed completely. Thus, RigL results in fewer, but more densely connected neurons, whereas the fan-in constraint enforces that all neurons are retained.

In Table 3 we compare SRigL to a variety of DST algorithms. SRigL performs comparably to other methods, even those which learn unstructured sparsity. Methods with a memory footprint listed as dense require training with the dense network and therefore are not directly comparable to other sparse-to-sparse DST methods. The most directly comparable method to ours is DSB; we note that SRigL outperforms DSB at all sparsity ratios reviewed.

**Table 1: Top-1 ImageNet test accuracy of ResNet-50** trained with RigL or SRigL at high sparsities and with various training times (as in Evci et al. (2021)), e.g. 5× more training epochs than dense ResNet-50.

| sparsity | RigL | | SRigL | | | |
| | | | w/o | w/ ablation | | |
| (%) | 1× | 5×[†] | 1× | 1× | 2× | 5× |
| --- | --- | --- | --- | --- | --- | --- |
| 80 | 74.9 | 77.1 | 74.8 | 75.0 | 76.5 | 77.2 |
| 90 | 72.8 | 76.6 | 72.6 | 72.7 | 74.7 | 76.2 |
| 95 | 69.6 | 74.6 | 68.8 | 69.1 | 71.5 | 73.6 |
| 99 | 51.4 | 61.9[‡] | 48.7 | 51.5 | 55.3 | 59.0 |
| 0 | *dense ResNet-50*: | 76.7 | | | | |

[†] 5× RigL results are from Evci et al. (2021)
[‡] uses a dense first layer, unlike other results

**Table 2: Test accuracy for ResNet-18 on CIFAR-10** trained with RigL or SRigL with/without neuron ablation at varying sparsities repeated with five different random seeds.

| sparsity | RigL | SRigL | |
| | | w/o | w/ ablation |
| (%) | | | |
| --- | --- | --- | --- |
| 80 | 95.2±0.1 | 95.2±0.1 | 95.2±0.0 |
| 90 | 95.1±0.1 | 95.0±0.1 | 95.1±0.1 |
| 95 | 94.6±0.2 | 94.5±0.3 | 94.7±0.2 |
| 99 | 92.9±0.1 | 91.5±0.3 | 92.8±0.1 |
| 0 | *dense ResNet-18*: | 95.5 | |

**Table 3: Top-1 ImageNet test accuracy of ResNet-50 trained with a variety of DST methods**, highlighting methods that both are sparse-to-sparse (i.e. sparse training) *and* learn structured sparsity similar to SRigL — only DSB-16 (2:4 and 1:4 sparsity) is directly comparable in this regard. RigL and SRigL results are from our experiments, other values are obtained from each method's corresponding paper, unless noted otherwise.

| | training | | sparsity | | | | |
| method | method | structured | 50% | 75% | 80% | 90% | 93.75% |
| --- | --- | --- | --- | --- | --- | --- | --- |
| Static* | sparse | no | – | – | 70.6±0.06 | 65.8±0.04 | – |
| SET* | sparse | no | – | – | 72.9±0.39 | 69.6±0.23 | – |
| DeepR[§] | sparse | no | – | – | 71.7 | 70.2 | – |
| DSR | sparse | no | – | – | 73.3 | 71.6 | – |
| Top-KAST[‡] | sparse | no | – | – | 74.76 | 70.42 | – |
| MEST[†] | sparse | no | – | – | **75.39** | 72.58 | – |
| RigL | sparse | no | – | – | 74.98 | 72.81 | – |
| DSB-16 | **sparse** | **yes** | 76.33 | 74.04 | – | – | – |
| Chase[††] | **sparse** | **yes** | – | – | 75.27 | **74.03** | – |
| SRigL (Ours) | **sparse** | **yes** | **76.60** | **75.55** | 75.01 | 72.71 | 70.56 |
| SNFS (ERK)* | dense | no | – | – | 75.2±0.11 | 73.0±0.04 | – |
| AST+GC** | dense | no | – | – | 73.2 | 73.1 | – |
| SR-STE | dense | yes | – | 76.2 | – | – | 71.5 |
| | *dense ResNet-50:* | | | | 76.7 | | |

*Values obtained from Evci et al. (2021). §values obtained from Mostafa & Wang (2019). †Values for the MEST (x0.67+EM) variant, matched to the same number of training FLOPs as RigL. ‡Values tabulated for Top-KAST correspond to the *backwards sparsity* as Top-KAST uses different sparsities in the forward and backward passes. For more information see Table 1 in Jayakumar et al. (2020). ††Values from Yin et al. (2023) for channel sparsity ($S_c$) set to 40%. **50% initial sparsity. Values from Yang et al. (2022)

**Table 4: Top-1 test accuracy of ViT-B/16** trained on ImageNet with or w/o neuron ablation

| | RigL | SRigL | |
| sparsity (%)[†] | | w/o | w/ ablation |
| --- | --- | --- | --- |
| 80 | **77.9** | 73.5 | 77.5 |
| 90 | **76.4** | 71.3 | 76.0 |
| 0 | *dense ViT-B/16*: 78.35 | | |

†Sparsity level set for all modules *except multi-headed attention input projections*, which remain dense. See Appendix D.3 for more details.

**Table 5: SRigL sparsity and FLOPs** for ResNet-50/ImageNet training and inference. See Appendix G for more details.

| | SRigL FLOPs | |
| sparsity (%) | training ($\times 1e18$) | inference ($\times 1e9$) |
| --- | --- | --- |
| 80 | 1.13 | 3.40 |
| 90 | 0.77 | 1.99 |
| 95 | 0.40 | 1.01 |
| 99 | 0.09 | 0.21 |
| 0 | 3.15 | 8.20 |

### 4.3 VISION TRANSFORMER TRAINED ON IMAGENET

We train the vision transformer variant ViT-B/16 on ImageNet generally following the original training recipe per Dosovitskiy et al. (2021) with select modifications, see Appendix D.3 for more information.

Similar to our Convolutional Neural Network (CNN) experiments, RigL ablates a significant number of neurons when applied to the ViT-B/16 architecture with sparsities of 80 and 90%. Additionally, we find that RigL learns sparse connectivities with a high variance of fan-in between neurons (see Fig. 12). At 90% sparsity, some neurons are allocated up to $\times 10$ more active weights than the mean number of active weights in the same layer. We hypothesize that these more densely connected neurons found in our RigL experiments are important for generalization performance; therefore, a high $\gamma_{sal}$ threshold should improve performance of SRigL by ablating neurons until a sufficient density of sparse fan-in is reached. Indeed, we find that SRigL's generalization performance is sensitive to $\gamma_{sal}$ and that high $\gamma_{sal}$ thresholds of 90% to 99% perform best. See Fig. 9a and Appendix E for more details on how $\gamma_{sal}$ affects the generalization performance of ViT-B/16. For the following results, we used a $\gamma_{sal}$ of 95%.

We train each model with a single random initialization and report the results in Table 4. SRigL without ablation is unable to match the generalization performance of RigL at very high sparsity. However, with neuron ablation enabled, SRigL's performance greatly improves and is closely comparable to RigL at 80% and 90% sparsity.

### 4.4 ACCELERATION OF CONSTANT FAN-IN SPARSITY

---

**Algorithm 1** "Condensed" linear layer with constant fan-in sparsity forward pass

---

```
 1: Input: x: the input matrix of shape (batch_size, num_features)
 2:        w: the condensed weight matrix of shape (active_neurons, constant_fan_in)
 3:        indx: indices of non-zero dense weights of shape (active_neurons,
 4:        constant_fan_in)
 5: output ← torch.zeros(size=(batch_size, neurons))
 6: for b in range(batch_size) do              ▷ For each sample in mini-batch
 7:     for n in range(neurons) do             ▷ For each active neuron in layer
 8:         for k in range(constant_fan_in) do        ▷ For each non-zero weight
 9:             source_idx ← idx[n, k]
10:             feature ← x[b, source_idx]
11:             output[b, n] += feature * w[n, k]
12: return output
```

---

While SRigL shows promising theoretical speedups (i.e. FLOPs) as demonstrated in Table 5 and Appendix G, FLOPs are limited in demonstrating the real-world acceleration potential of a proposed sparse representation in general. Yet conversely, creating a fully-optimized software or hardware implementation of a novel representation typically requires significant engineering effort outside of the scope of this paper.

Here we show that even a straight-forward PyTorch implementation of our proposed condensed neural network representation (see Appendix F) can demonstrate this real-world acceleration. The algorithm to accelerate our condensed sparsity representation is shown in Algorithm 1, demonstrating that it is embarrassingly parallel. Additionally, leveraging CUDA kernels from Schultheis & Babbar (2023), we also demonstrate that constant fan-in sparsity can be accelerated on commodity GPUs.

To accelerate our condensed linear layer we exploit both structured and constant fan-in sparsity by removing ablated neurons and zero-valued weights from active neurons. In Fig. 4, we present real-world timings comparing our condensed linear layer to structured and unstructured sparse representations. We extract the trained layer weights and bias from ViT-B/16 models trained with SRigL to obtain an accurate representation of the sparse topology produced during a real training run with SRigL.

Our condensed representation is significantly faster than the dense benchmark and other sparse representations across all sparsities investigated. This real-world speed-up is immediately applicable to applications where latency is critical. In some instances, we found structured sparsity yields the best acceleration. By including both structured and constant fan-in sparsity, models trained with SRigL can use either the fully condensed (structured + constant fan-in) *or* purely structured sparse representations to obtain real-world acceleration across a broad range of applications with the *same set of weights*.

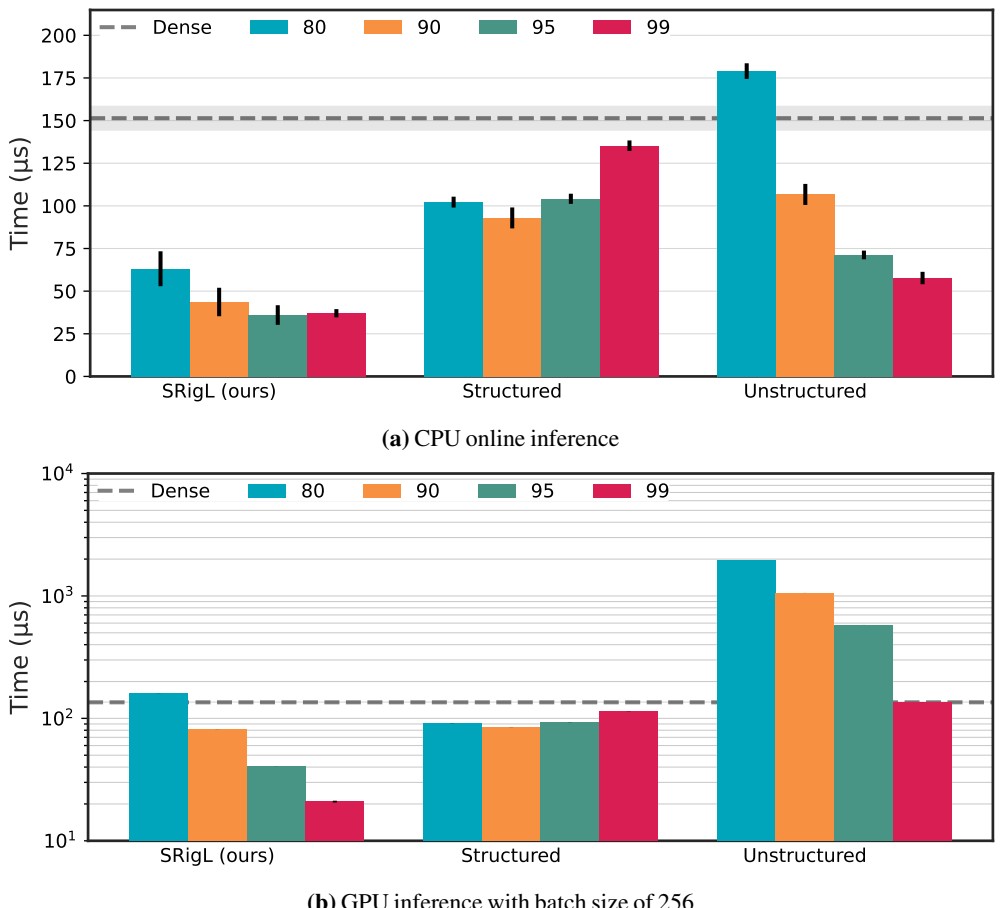

**(a)** CPU online inference

**(b)** GPU inference with batch size of 256

**Figure 4: Comparing real-world timings for a fully-connected layer** extracted from a ViT-B/16 model trained with SRigL when compressed using the condensed representation learned by SRigL to structured (i.e. the same layer accelerated using only the ablated neurons without exploiting the fine-grained sparsity), and unstructured (i.e. Compressed Sparse Row (CSR)) representations. The median over a minimum of 5 runs is shown, while the error bars show the std. dev. *Note: the increased timings for the 95 & 99% sparse structured representations is due to SRigL ablating relatively fewer neurons at these sparsities compared to 80 and 90%.* **(a)** **CPU wall-clock timings for online inference** on an Intel Xeon W-2145. For online (single input) inference, our condensed representation at 90% is *3.4× faster than dense* and *2.5 × faster than unstructured sparsity*. See Appendix I. **(b)** **GPU wall-clock timings for inference with a batch size of 256** on an NVIDIA Titan V. At 90% sparsity, our condensed representation is *1.7× faster than dense* and *13.0× faster than unstructured (CSR) sparse layers.* Note y-axis is log-scaled.

See Appendix I and Appendix J for details on wall-clock benchmarks across a range of threads and batch sizes. Furthermore, we expect that a more optimized software implementation and/or explicit hardware support would enable use of SRigL across a wider range of applications.

## 5 CONCLUSION

In this work we present SRigL, a novel DST method that learns a sparsity mask incorporating both structured and constant fan-in sparsity. SRigL is capable of sparse-to-sparse training while maintaining generalization performance on par with state-of-the-art unstructured sparse training methods on a wide variety of network architectures. Our observation that RigL ablates neurons at high sparsities inspires our neuron ablation method which enables SRigL to match the performance of RigL, even at high sparsities and on the ViT-B/16 network architecture. SRigL's constant fan-in constraint and neuron ablation results in real-world acceleration for CPU online inference and GPU batched inference. We hope this work will motivate the implementation of additional fine-grained structured sparsity schemes and the engineering efforts required to accelerate them further.

ACKNOWLEDGMENTS

We acknowledge the support of Alberta Innovates, the Natural Sciences and Engineering Research Council of Canada (NSERC), and the NSF AI Institute for Artificial Intelligence and Fundamental Interactions (IAIFI). We are grateful for computational resources made available to us by Denvr Dataworks, Google, Amazon, and the Digital Research Alliance of Canada. We also acknowledge the very helpful feedback of Erik Schultheis and Trevor Gale.

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

# Appendices

## A    SPARSITY AND OUTPUT-NORM VARIANCE

Consider a SNN with ReLU activations, where each neuron has on average $k$ connections to the previous layer (i.e., fan-in). It has been shown by Evci et al. (2022), that by normalizing the weights on initialization by a factor of $\sqrt{2/k}$, one achieves the following desirable normalization property for each layer $\ell$ with output $z^\ell$:

$$\mathbb{E}\left(\frac{||z^{\ell+1}||^2}{||z^\ell||^2}\right) = 1,$$

Meaning that on average the variance of the norm of each layer's output is constant. However, the variance of this ratio is non-trivial. In networks with large depth, it can accumulate, leading to exponentially large variance at the final layer (Li et al., 2021). Minimizing this variance on initialization has been shown to have a positive effect on training dynamics in some network models (Littwin et al., 2020), as it stabilizes the gradients. We therefore analyze the output norm variance as a guiding quantity for sparsity-type selection.

In the following, we consider three different types of sparsity distributions, which respectively correspond to different degrees of sparsity *structure* in the SNN, and derive analytic expressions for the behaviour of output norm variance in SNNs with the given sparsity type. The derivations for the following results can be found in Appendix B:

- **"Bernoulli sparsity"**: A connection between each neuron in layer $\ell+1$ and each neuron in layer $\ell$ appears *independently* with probability $p = \frac{k}{n}$, resulting in each neuron having $k$ connections *on average* and each layer having $nk$ connections *on average*. The variance is:

$$\mathbf{Var}_{\text{Bernoulli}}\left(\frac{||z^{\ell+1}||^2}{||z^\ell||^2}\right) = \frac{5n-8+18\frac{k}{n}}{n(n+2)}. \tag{1}$$

- **"Constant Per-Layer sparsity"**: Exactly $kn$ connections are distributed at random in the layer connecting the $n$ neurons in layer $\ell+1$ and the $n$ neurons in layer $\ell$, resulting in each neuron having $k$ connections *on average*. The variance is:

$$\mathbf{Var}_{\text{Const-Per-Layer}}\left(\frac{||z^{\ell+1}||^2}{||z^\ell||^2}\right) = \frac{(n^2+7n-8)C_{n,k}+18\frac{k}{n}-n^2-2n}{n(n+2)}, \tag{2}$$

where $C_{n,k} = \frac{n-1/k}{n-1/n}$. Note that when $n \gg 1$, $C_{n,k} \approx 1 - \frac{n-k}{n^2 k}$ is close to 1, and with $C_{n,k} = 1$ we recover the formula for Bernoulli sparsity, meaning that this sparsity type and Bernoulli sparsity are very similar.

- **"Constant Fan-In sparsity"**: Each neuron in layer $\ell+1$ is connected to exactly $k$ neurons from layer $\ell$, chosen uniformly at random. In this case, the variance is:

$$\mathbf{Var}_{\text{Const-Fan-In}}\left(\frac{||z^{\ell+1}||^2}{||z^\ell||^2}\right) = \frac{5n-8+18\frac{k}{n}}{n(n+2)} - \frac{3(n-k)}{kn(n+2)}. \tag{3}$$

In deriving the above results we assumed that the direction of the layer output vector $\frac{z^\ell}{||z^\ell||}$ is uniformly distributed on the unit sphere. We compare our theoretical predictions with simulations in Fig. 1b and verify their accuracy. Bernoulli and constant-per-layer distribution result in unstructured sparsity, and most of the current DST approaches, including RigL, operate with constant-per-layer sparsity. In contrast, the constant-fan-in type imposes a strong structural constraint. Therefore we are somewhat surprised to find that, in fact, constant-fan-in sparsity always produces slightly smaller output-norm variance than the other types. The difference is larger when $k \ll n$, i.e., for very sparse networks. This indicates that, at the very least, the constant fan-in constraint should not impair SNN training dynamics and performance, motivating our method of maintaining the constant fan-in sparsity constraint within a DST approach.

## B  COMPUTING THE OUTPUT NORM VARIANCE

**Definition B.1.** *Let $\xi \in \{0,1\}^N$ be a binary vector. Let $I \in \{0,1\}^{N \times N}$ be an $N \times N$ binary matrix. Let $u \in \mathbb{R}^N$ be any vector. Let $W \in \mathbb{R}^{N \times N}$ be a matrix of iid $\mathcal{N}(0,1)$ random variables.*

Define the vector $z$ by:

$$z = \sqrt{\frac{2}{k}}(W \odot I)(\xi \odot u) \tag{4}$$

i.e. the entries $z_i$ are given by:

$$z_i = \sqrt{\frac{2}{k}} \sum_{j=1}^n W_{ij} I_{ij} \xi_j u_j \tag{5}$$

**Proposition B.2.** *The variance of each entry $z_i$ is:*

$$\mathbf{Var}(z_i) = \frac{2}{k} \sum_{j=1}^n I_{ij} \xi_j u_j^2 \tag{6}$$

*and therefore the distribution of each $z_i$ can be written as*

$$z_i \overset{d}{=} g_i \sqrt{\frac{2}{k} \sum_{j=1}^n I_{ij} \xi_j u_j^2} \tag{7}$$

*where $g_i$ are $N$ iid $\mathcal{N}(0,1)$ random variables.*

*Proof.* By the properties of variance:

$$\mathbf{Var}(z_i) = \frac{2}{k} \sum_{j,j'} I_{ij} I_{ij'} \xi_j \xi_{j'} u_j u_j' \mathbf{Cov}(W_{ij}, W_{ij'}) \tag{8}$$

$$= \frac{2}{k} \sum_{j,j'} I_{ij} I_{ij'} \xi_j \xi_{j'} u_j u_j' \delta_{j=j'} \tag{9}$$

$$= \frac{2}{k} \sum_j I_{ij}^2 \xi_j^2 u_j^2 \tag{10}$$

$$= \frac{2}{k} \sum_j I_{ij} \xi_j u_j^2 \tag{11}$$

since $I_{ij}^2 = I_{ij}$ and $\xi_j^2 = \xi_j$ because they are binary valued. Once the variance is established, notice that $z_i$ is a linear combination of Gaussians with $z_i \perp z_{i'}$, because the row $W_{ij} \perp W_{i'j}$. Hence the $z_i$ are independent Gaussians, so the form $z_i \overset{d}{=} g_i \sqrt{\frac{2}{k} \sum_{j=1}^n I_{ij} \xi_j u_j^2}$ follows. $\square$

**Corollary B.3.** *The norm $\|z\|^2$ can be written as:*

$$\|z\|^2 \overset{d}{=} \frac{2}{k} \sum_{i,j=1}^n g_i^2 I_{ij} \xi_j u_j^2 \tag{12}$$

**Proposition B.4** ("Bernoulli Sparsity")**.** *Suppose that $u \in \mathbb{R}^n$ is uniform from the unit sphere, the entries $I_{ij} \sim Ber\left(\frac{k}{n}\right)$, $\xi_j \sim Ber\left(\frac{1}{2}\right)$ all independent of each other. Then:*

$$\mathbb{E}\left(\|z\|^2\right) = 1 \tag{13}$$

$$\mathbf{Var}\left(\|z\|^2\right) = \frac{5n - 8 + 18\frac{n}{k}}{n(n+2)} \tag{14}$$

| Case | Num. Terms | $\mathbb{E}\big[g_i^2 g_{i'}^2\big]$ | $\mathbb{E}[I_{i'j'}I_{ij}]$ | $\mathbb{E}[\xi_j\xi_{j'}]$ | $\mathbb{E}\big[u_j^2 u_{j'}^2\big]$ |
|---|---|---|---|---|---|
| $i=i',j=j'$ | $n^2$ | 3 | $\frac{k}{n}$ | $\frac{1}{2}$ | $\frac{3}{n(n+2)}$ |
| $i\neq i',j=j'$ | $n^2(n-1)$ | 1 | $\left(\frac{k}{n}\right)^2$ | $\frac{1}{2}$ | $\frac{3}{n(n+2)}$ |
| $i=i',j\neq j'$ | $n^2(n-1)$ | 3 | $\left(\frac{k}{n}\right)^2$ | $\left(\frac{1}{2}\right)^2$ | $\frac{1}{n(n+2)}$ |
| $i\neq i',j\neq j'$ | $n^2(n-1)^2$ | 1 | $\left(\frac{k}{n}\right)^2$ | $\left(\frac{1}{2}\right)^2$ | $\frac{1}{n(n+2)}$ |

**Table 6:** Overview of terms for Bernoulli type sparsity.

*Proof.* We have

$$\mathbb{E}\left(\|z\|^2\right)=\frac{2}{k}\sum_{i,j=1}^{n}\mathbb{E}\big[g_i^2 I_{ij}\xi_j u_j^2\big] \tag{15}$$

$$=\frac{2}{k}\sum_{i,j=1}^{n}\mathbb{E}\big[g_i^2\big]\mathbb{E}[I_{ij}]\mathbb{E}[\xi_j]\mathbb{E}\big[u_j^2\big] \tag{16}$$

$$=\frac{2}{k}\sum_{i,j=1}^{n}1\cdot\frac{k}{n}\cdot\frac{1}{2}\cdot\frac{1}{n} \tag{17}$$

$$=1 \tag{18}$$

Similarly, we compute the 4-th moment as follows:

$$\mathbb{E}\left(\|z\|^4\right)=\left(\frac{2}{k}\right)^2\sum_{i,j,i',j'}^{n}\mathbb{E}\big[g_i^2 g_{i'}^2\big]\mathbb{E}[I_{i'j'}I_{ij}]\mathbb{E}[\xi_j\xi_{j'}]\mathbb{E}\big[u_j^2 u_{j'}^2\big] \tag{19}$$

We split this into four cases and evaluate these based on whether or not $i=i'$ and $j=j'$ in the following table.

Combining the value of each term with the number of terms gives the desired result for the variance. $\quad\square$

**Proposition B.5** ("Constant-per-layer sparsity"). *Suppose that $u\in\mathbb{R}^n$ is uniform from the unit sphere and $\xi_j\sim Ber(\frac{1}{2})$ are independent of each other. Suppose the entries of the matrix $I_{ij}$ are chosen such that:*

*There are exactly $kn$ ones and exactly $n^2-nk$ zeros in the matrix $I$, and their positions in the matrix are chosen uniformly from the $\binom{n^2}{nk}$ possible configurations. Then:*

$$\mathbb{E}\left(\|z\|^2\right)=1 \tag{20}$$

$$\mathbf{Var}\left(\|z\|^2\right)=\frac{(n^2+7n-8)C_{n,k}+18\frac{k}{n}-n^2-2n}{n(n+2)} \tag{21}$$

*Proof.* Note that $\mathbb{E}(I_{ij})=k/n$ still holds, since there are $kn$ ones distributed over $n^2$ locations. Thus the computation for $\mathbb{E}(\|z\|^2)$ is identical to the previous proposition. Note also that when there are two entries, we have:

$$\mathbb{E}[I_{ij}I_{i'j'}]=\begin{cases}\frac{k}{n} & \text{if } i=i' \text{ and } j=j'\\ \frac{k}{n}\cdot\frac{nk-1}{n^2-1} & \text{otherwise}\end{cases} \tag{22}$$

$$=\begin{cases}\frac{k}{n} & \text{if } i=i' \text{ and } j=j'\\ \left(\frac{k}{n}\right)^2\cdot C_{n,k} & \text{otherwise}\end{cases} \tag{23}$$

where $C_{n,k}=\frac{n-1/k}{n-1/n}$. The table with terms for computing $\mathbb{E}(\|z\|^4)$ becomes: The extra factor of $C_{n,k}$

in the entries leads to the stated result. $\quad\square$

| Case | Num. Terms | $\mathbb{E}\big[g_i^2 g_{i'}^2\big]$ | $\mathbb{E}[I_{i'j'}I_{ij}]$ | $\mathbb{E}[\xi_j \xi_{j'}]$ | $\mathbb{E}\big[u_j^2 u_{j'}^2\big]$ |
|---|---|---|---|---|---|
| $i=i', j=j'$ | $n^2$ | 3 | $\frac{k}{n}$ | $\frac{1}{2}$ | $\frac{3}{n(n+2)}$ |
| $i\neq i', j=j'$ | $n^2(n-1)$ | 1 | $\left(\frac{k}{n}\right)^2 C_{n,k}$ | $\frac{1}{2}$ | $\frac{3}{n(n+2)}$ |
| $i=i', j\neq j'$ | $n^2(n-1)$ | 3 | $\left(\frac{k}{n}\right)^2 C_{n,k}$ | $\left(\frac{1}{2}\right)^2$ | $\frac{1}{n(n+2)}$ |
| $i\neq i', j\neq j'$ | $n^2(n-1)^2$ | 1 | $\left(\frac{k}{n}\right)^2 C_{n,k}$ | $\left(\frac{1}{2}\right)^2$ | $\frac{1}{n(n+2)}$ |

**Table 7:** Overview of terms for Constant-per-layer type sparsity.

**Proposition B.6** ("Constant Fan-In sparsity"). *Suppose that $u \in \mathbb{R}^n$ is uniform from the unit sphere, and $\xi_j \sim Ber\left(\frac{1}{2}\right)$ all independent of each other. Suppose the entries of the matrix $I_{ij}$ are chosen so that:*

1. *There are exactly $k$ ones **in each row** of the matrix $I$ and exactly $n-k$ zeros in the matrix $I$, chosen uniformly from the $\binom{n}{k}$ possible ways this can happen.*
2. *Different rows of $I$ are independent.*

*Then:*

$$\mathbb{E}\left(\|z\|^2\right) = 1 \tag{24}$$

$$\mathbf{Var}\left(\|z\|^2\right) = \frac{5n - 8 + 18\frac{n}{k}}{n(n+2)} - \frac{3(n-k)}{kn(n+2)} \tag{25}$$

*Proof.* Same arguments as before apply, but now we have

$$\mathbb{E}[I_{ij}I_{i'j'}] = \begin{cases} \frac{k}{n} & \text{if } i=i' \text{ and } j=j' \\ \frac{k}{n}\frac{k-1}{n-1} & \text{if } i=i' \text{ and } j\neq j' \\ \left(\frac{n}{n}\right)^2 & \text{otherwise} \end{cases} \tag{26}$$

$$\tag{27}$$

and the table for the variance computation becomes:

| Case | Num. Terms | $\mathbb{E}\big[g_i^2 g_{i'}^2\big]$ | $\mathbb{E}[I_{i'j'}I_{ij}]$ | $\mathbb{E}[\xi_j \xi_{j'}]$ | $\mathbb{E}\big[u_j^2 u_{j'}^2\big]$ |
|---|---|---|---|---|---|
| $i=i', j=j'$ | $n^2$ | 3 | $\frac{k}{n}$ | $\frac{1}{2}$ | $\frac{3}{n(n+2)}$ |
| $i\neq i', j=j'$ | $n^2(n-1)$ | 1 | $\left(\frac{k}{n}\right)^2$ | $\frac{1}{2}$ | $\frac{3}{n(n+2)}$ |
| $i=i', j\neq j'$ | $n^2(n-1)$ | 3 | $\frac{k}{n}\cdot\frac{k-1}{n-1}$ | $\left(\frac{1}{2}\right)^2$ | $\frac{1}{n(n+2)}$ |
| $i\neq i', j\neq j'$ | $n^2(n-1)^2$ | 1 | $\left(\frac{k}{n}\right)^2$ | $\left(\frac{1}{2}\right)^2$ | $\frac{1}{n(n+2)}$ |

**Table 8:** Overview of terms for Constant-fan-in type sparsity.

Which leads to the stated result. $\qquad\square$

$$\tag{28}$$

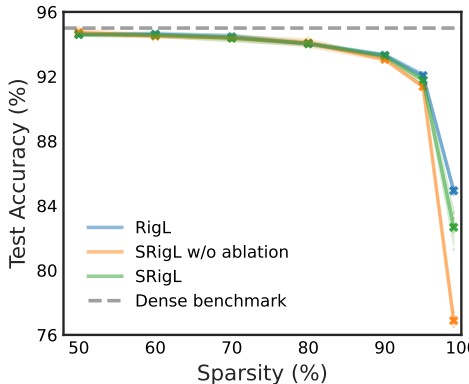

| | RigL | SRigL | |
|---|---|---|---|
| sparsity (%) | | w/o | w/ ablation |
| 50 | 94.6±0.1 | 94.7±0.1 | 94.6±0.1 |
| 60 | 94.6±0.1 | 94.5±0.1 | 94.6±0.1 |
| 70 | 94.5±0.1 | 94.4±0.1 | 94.4±0.1 |
| 80 | 94.0±0.1 | 94.1±0.2 | 94.0±0.1 |
| 90 | 93.3±0.1 | 93.1±0.1 | 93.3±0.1 |
| 95 | 92.1±0.1 | 91.4±0.1 | 91.8±0.2 |
| 99 | **84.9±0.2** | 76.9±0.3 | 82.7±0.8 |
| 0 | *dense Wide ResNet-22*: | 95.0 | |

**Figure 5 & Table 9:** Test accuracy of Wide ResNet-22 trained on CIFAR-10. Mean and 95% confidence intervals are reported over five runs.

## C  WIDE RESNET-22 TRAINED ON CIFAR-10

In Fig. 5 we present results of training Wide ResNet-22 (Zagoruyko & Komodakis, 2017) with RigL or SRigL on the CIFAR-10 dataset. The training details for this experiment are identical to those reported in Section 4.1. SRigL without ablation performs poorly at very high sparsities. With ablation, SRigL achieves generalization performance comparable to RigL.

## D  HYPERPARAMETER AND TRAINING DETAILS

### D.1  RESNET-18 TRAINED ON CIFAR-10

As per Liu (2017), we modify the original ResNet-18 network by changing the kernel dimensions of the first convolutional layer to $3\times3$ instead of $7\times7$. Further, we reduce the stride in the first two convolutional layers to one to avoid excessive reduction of the feature map's spatial dimensions.

We train each network for 250 epochs (97,656 steps) using a batch size of 128. An initial learning rate of 0.1 is reduced by a factor of 5 every 77 epochs (about 30,000 steps). We use stochastic gradient descent (SGD) with momentum, with an L2 weight decay coefficient of 5e-4 and momentum coefficient of 0.9. We train each model using a single Nvidia V100 GPU.

We achieve the desired overall sparsity by distributing the per-layer sparsity according to the ERK (Evci et al., 2021; Mocanu et al., 2018) distribution, which scales the per-layer sparsity based on the number of neurons and the dimensions of the convolutional kernel, if present. We set the number of mini-batch steps between connectivity updates, $\Delta T$, to 100. $\gamma_{sal}$ is set at 30% based on the results of a small grid search performed on CIFAR-10 with ResNet-18 and Wide ResNet-22. See Fig. 8 for details.

For each trial, we select a desired sparsity in the range from 0.5 to 0.99. At each connectivity update, the portion of weights to be pruned or regrown is based on a cosine annealing schedule (Dettmers & Zettlemoyer, 2019) with an initial value $\alpha=0.3$. The portion of weights to be updated decays from the initial value to zero once 75% of the total training steps have been completed, after which the weight mask remains constant.

### D.2  RESNET-50 TRAINED ON IMAGENET

We use a mini-batch size of 512 instead of 4096, We linearly scale the learning rate and $\Delta T$ to account for our smaller batch size. Linearly scaling the learning rate in this manner was included in the original RigL source code and is further motivated by Goyal et al. (2018). We increase $\Delta T$ to 800 and average the dense gradients over eight mini-batch steps to ensure that SRigL has the same quality of parameter saliency information available as RigL at each network connectivity update. We set $\gamma_{sal}$ to 30% based on our grid search presented in Fig. 8.

Our learning rate uses a linear warm-up to reach a maximum value of 0.2 at epoch five and is reduced by a factor of 10 at epochs 30, 70, and 90. Using a mini-batch of 512, we train the networks for 256,000 steps to match RigL's training duration. We use a cosine connectivity update schedule with $\alpha = 0.3$. We initialize the sparse model weights per Evci et al. (2022). We train the networks using SGD with momentum, L2 weight decay, and label smoothing (Szegedy et al., 2016) coefficients of 0.9, 1e-4 and 0.1, respectively.

We use the same standard data augmentation in our data preprocessing as RigL, including randomly resizing to 256×256 or 480×480 pixels, random crops to 224×224 pixels, random horizontal flips, and per-image normalization to zero mean and unit variance using identical per RGB channel mean and standard deviation values as RigL. We train each model using either four Nvidia V100 or A100 GPUs.

### D.3 VISION TRANSFORMER TRAINED ON IMAGENET

For our ViT-B/16 experiments, we used sparsity on the convolutional projection (input projection to patches), the fully connected layers in the feed forward (MLP) blocks and the output projections of the multi-headed attention (MHA) modules. We performed a lightweight ablation study on four ViT-B/16 networks trained on ImageNet to determine the affect of sparsifying the first convolutional projection layer as well as the input projection layers in the multi-headed attention modules. Based on the results of our ablation study, *we did not use sparsity on the MHA input projection layers or the scaled-dot products*. See Fig. 9b for more details. This setup is similar to the "Sparse FF" models investigated by Jaszczur et al. (2021). The global model sparsity level reported in Table 4 is calculated based on the sparse modules only. If we also consider the parameters in the MHA input projections as part of our parameter budget, the global model sparsities tabulated in Table 4 correspond to 60.35% and 67.90% for the rows labelled 80% and 90% sparsity, respectively.

We add additional data augmentations following the standard TorchVision (Maintainers & Contributors, 2016) ViT-B/16 training procedure for ImageNet. These data augmentations applied include: random cropping, resizing the cropped image to 224 by 224 pixels, randomly horizontal flips, randomly augmenting with RandAugment algorithm (Cubuk et al., 2020), and normalizing with the typical RGB channel mean and standard deviation values. We also randomly choose one of random mixup (Zhang et al., 2023) or random cutmix (Yun et al., 2019) and add it to the above-noted augmentations. We use 0.2 and 1.0 for the alpha parameter values for mixup and cutmix, respectively. We omit Dropout (Srivastava et al., 2014) from the model entirely to avoid potential layer collapse in the case where all non-zero weights are dropped from a layer and to avoid any other unintended interference with SRigL's sparse training procedure.

We sample eight mini-batch steps with 512 samples per mini-batch and accumulate gradients before applying the optimizer, resulting in an effective mini-batch size of 4096. We train the model for 150 epochs using an AdamW (Loshchilov & Hutter, 2018) optimizer with weight decay, label smoothing, $\beta_1$, and $\beta_2$ coefficients of 0.3, 0.11, 0.9, and 0.999, respectively. We use cosine annealing with linear warm-up for our learning rate scheduler with an initial learning rate of 9.9e-5 that warms-up to a maximum value of 0.003 at epoch 16. We clip all parameter gradients to a max L2 norm of 1.0. We apply uniformly distributed sparsity across all layers in the model. $\Delta T$ is set to 100 to update network connectivity every 100 mini-batch steps. We train each model using either four Nvidia V100 or A100 GPUs.

### D.4 MOBILENET-V3 TRAINED ON IMAGENET

We follow the TorchVision (Maintainers & Contributors, 2016) training recipe for MobileNet-V3 Large and Small for ImageNet. We set $\Delta T$ to 100 and $\gamma_{sal}$ to 30% similar to our other CNN experiments. We train the models from scratch for 600 epochs using an RMSProp (Tieleman et al., 2012) optimizer with a momentum, L2 weight decay, and smoothing constant coefficients of 0.9, 1e-5, and 0.9, respectively. The networks are trained with a step learning rate decay schedule with initial learning rate of 0.064, multiplicative factor of 0.973, and we decay the learning rate every two epochs.

The input data is augmented with random cropping to 224 by 224 pixels, random horizontal flips, AutoAugmentation using the ImageNet policy (Cubuk et al., 2019), normalizing to standard RGB mean and standard deviation values, and random erasing with a probability of 0.2 (Zhong et al., 2017). Similar to the above, we omit Dropout (Srivastava et al., 2014) to avoid potential layer collapse. Unlike the TorchVision recipe, we *do not* average the trained parameters across the last three checkpoints that improved the top-1 accuracy. We train with a batch size of 512 and accumulate gradients across

two mini-batches, resulting in an effective mini-batch size of 1024. We train each model using four Nvidia A100 GPUs.

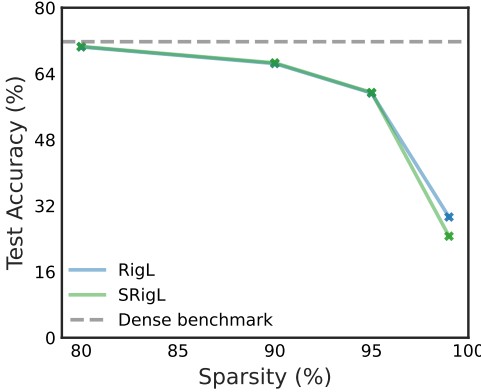 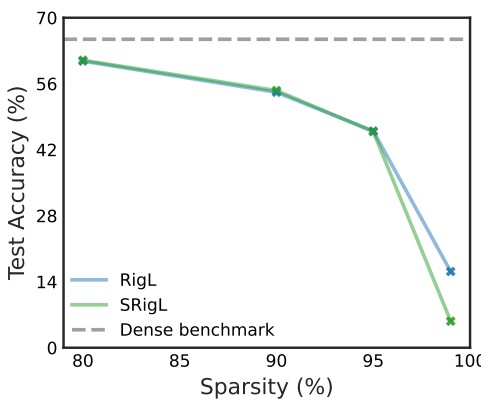

**Figure 6: MobileNet-V3 Large / ImageNet** top-1 accuracy. SRigL compares well against RigL both both models perform poorly compared to the denes baseline at 99% sparsity.

**Figure 7: MobileNet-V3 Small / ImageNet** top-1 accuracy. SRigL compares well against RigL both both models perform poorly compared to the denes baseline at 99% sparsity.

# E  TUNING $\gamma_{sal}$, MINIMUM PERCENTAGE SALIENT WEIGHTS PER NEURON

Fig. 8 depicts the generalization performance of highly sparse ResNet-18 and Wide ResNet-22 models trained on the CIFAR-10 dataset. SRigL's generalization performance at high sparsities is improved with neuron ablation; however, the specific value selected for $\gamma_{sal}$ does not have a significant effect on performance. Our experiments demonstrate that SRigL performs well with a variety of $\gamma_{sal}$ values. In Section 4 we report the results of SRigL models trained with $\gamma_{sal}$ set to 30%. With dynamic ablation enabled, we set the minimum salient weights per neuron to one if the user-defined threshold results in a value less than one. In Fig. 10, many layers in ResNet-50 are set to the minimum threshold of one when we apply a $\gamma_{sal}$ of 30% for all model types other than ViT-B/16. This minimum threshold explains the invariance of the model's performance when comparing against multiple values for $\gamma_{sal}$.

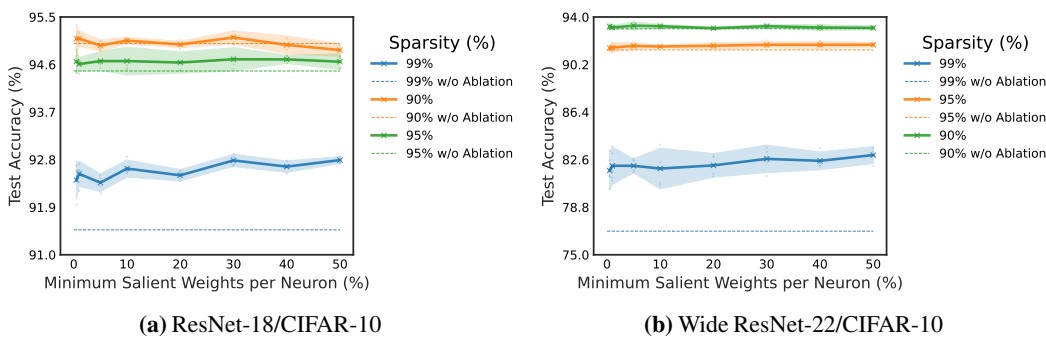

**(a)** ResNet-18/CIFAR-10          **(b)** Wide ResNet-22/CIFAR-10

**Figure 8: (a) ResNet18/CIFAR-10 Test Accuracy vs.** $\gamma_{sal}$ when trained with SRigL with and without ablation for a range of sparsities. The mean and 95% confidence intervals are shown for five different random seeds for the runs with ablation. For the runs without ablation, we report the mean of five different random seeds. **(b) Wide ResNet-22 Test Accuracy vs.** $\gamma_{sal}$. The mean and 95% confidence intervals are shown for five different random seeds.

Fig. 9a demonstrates how ViT-B/16's generalization performance is much more sensitive to $\gamma_{sal}$. We find that RigL learns a sparse connectivity pattern with a large variance in sparse fan-in between neurons within a given layer, with some neurons having an order of magnitude more fan-in connection than the mean fan-in.

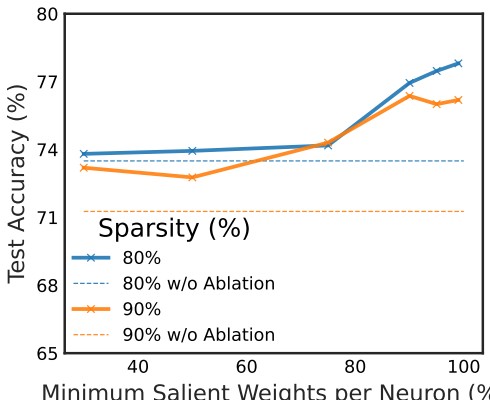 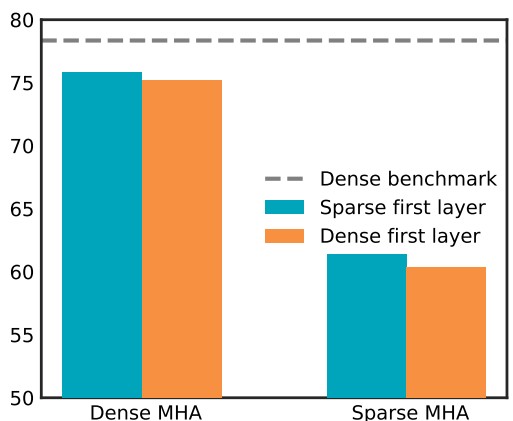

**(a) ViT-B/16/ImageNet Test Accuracy vs.** $\gamma_{sal}$ when trained with SRigL with and without ablation enabled for 80% and 90% sparsity. ViT-B/16's performance is much more sensitive to $\gamma_{sal}$ and generally performs best with high ablation thresholds. Based on this data we set $\gamma_{sal}$ to 95% for our results reported in Section 4.3.

**(b) ViT-B/16 ablation study**. The best performing variant used a sparse first layer and dense input projections in the MHA modules.

## F  CONDENSED MATRIX MULTIPLICATION

Using a constant fan-in sparse representation presents an advantage compared to the general N:M sparse representation in that we can represent our weight matrices in a compact form, since every neuron/convolutional filter has the same number of non-zero weights. Here we demonstrate how this can be used to accelerate a fully-connected layer.

Consider the standard matrix-vector product:

$$
Wv = \begin{pmatrix} W_{11} & W_{12} & \dots & W_{1d} \\ W_{21} & W_{22} & \dots & W_{2d} \\ \vdots & \vdots & \ddots & \vdots \\ W_{n1} & W_{n2} & \dots & W_{nd} \end{pmatrix} \begin{pmatrix} v_1 \\ v_2 \\ \vdots \\ v_d \end{pmatrix} = \begin{pmatrix} \sum_{j=1}^{d} W_{1j}v_j \\ \sum_{j=1}^{d} W_{2j}v_j \\ \vdots \\ \sum_{j=1}^{d} W_{nj}v_j \end{pmatrix} = v^{\text{out}} \tag{29}
$$

When $W \in \mathbb{R}^{n \times d}$ is sparse and has only $k$ non-zero elements per row, the sums representing each element of $v^{\text{out}}$ will be limited to $k$ terms, i.e.:

$$
v_i^{\text{out}} = \sum_{\alpha=1}^{k} W_{ij_\alpha} v_{j_\alpha} \quad \text{with } j_\alpha \in \{1,...,d\}, \quad j_\alpha \neq j_{\alpha'} \tag{30}
$$

Note that the expression on the right-hand side of Eq. (29) can be represented as an operation between a dense matrix $W^c \in \mathbb{R}^{n \times k}$ (we call it "condensed $W$") and $k$ vectors $v^{\pi_1},...,v^{\pi_k}$, $v^{\pi_i} \in \mathbb{R}^n$, whose elements are drawn from $v$ with replacement (we call them "recombinations of $v$"). The operation is a sum over element-wise products between the $i$-th column of $W^c$ and the $i$-th column vector $v^{\pi_i}$:

$$
Wv = \sum_{i=1}^{k} W_{:,i}^c \odot v^{\pi_i} \tag{31}
$$

Mathematically, these methods are equivalent for any matrices. Computationally, the condensed method can be more efficient, in particular for sparse matrices with constant small fan-in $k$. By construction, this method requires the sparse matrix $W$ to be stored in dense representation which involves two 2D arrays of shape $n \times k$: One holds the *values* of the non-zero elements of $W$ and the other one their respective *column indices*, which are used to generate input vector re-combinations. An efficient computational implementation of this method is subject of ongoing work on this project. Based on our results, the constant fan-in constraint does not appear to have a limiting effect on SNNs.

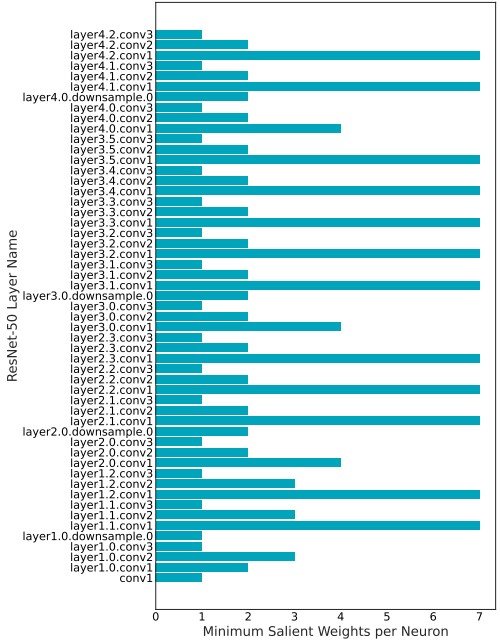

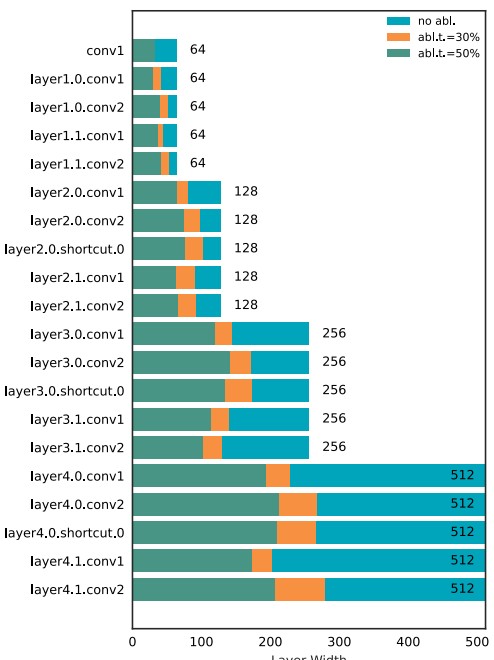

**Figure 10: ResNet-50 Layer vs. Minimum salient weights per neuron**. SRigL sets the minimum salient weight per neuron to 1 if the product between $\gamma_{sal}$ and the sparse fan-in per neuron is less than 1. Therefore, even in a relatively large network such as ResNet50 many of the layers only require that a single weight be active to keep the neuron active. We believe this is why SRigL's performance is relatively invariant to various ablation thresholds when applied to CNNs

**Figure 11: ResNet-18/CIFAR-10 layer widths at the end of training at 99% sparsity**. Without ablation, constant fan-in constraint enforces that sparse layers retain their original width. When ablation is enabled, the $\gamma_{sal}$ threshold (minimum percentage salient weights per neuron) is used to control the amount of ablation.

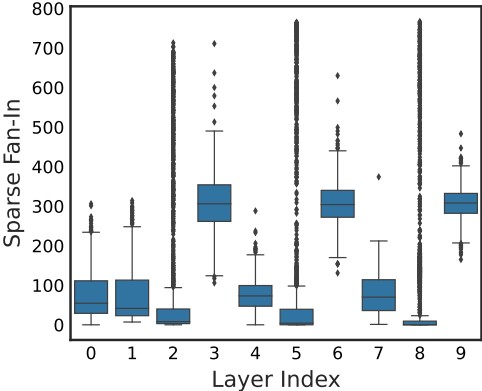

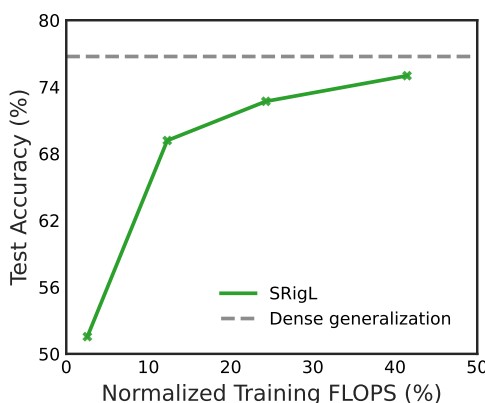

**Figure 12: Sparse Fan-In vs. ViT-B/16 layer index at the end of training with RigL at 90% sparsity**. Only the first 10 layers are shown for clarity. We find that RigL learns a sparse connectivity with large variance in fan-in between neurons within the same layer with some neurons receiving up to ×10 the number of active connections than the mean for the same layer.

**Figure 13: Training FLOPs** for SRigL on ResNet-50/ImageNet at a variety of sparsities compared with dense generalization. FLOPs are normalized by dense training FLOPs.

## G   FLOPs ANALYSIS

In Fig. 13, we present an analysis of the FLOPs required during training and inference for SRigL and compare with SR-STE. We calculate FLOPs using the same methodology as Evci et al. (2021) by considering only operations induced by convolutional and linear layers and their activations. FLOPs for add and pooling operations are ignored. For training FLOPs, we also disregard FLOPs required for mask updates, as this step is amortized over $\Delta T$ steps and is negligible compared to the FLOPs required otherwise for training. The open-source code for counting operations is from the NeurIPS 2019 MicroNet Challenge and is available on GitHub[2].

Similar to other DST methods, SRigL obtains generalization performance comparable to a dense network benchmark at a fraction of the FLOPs required for both training and inference.

## H   IN TIME OVERPARAMETERIZATION RATES

In Figs. 14 to 17 we present the In Time Overparameterization Rate (ITOP) (Liu et al., 2021c) for various models and datasets. In this same work, Liu et al. (2021c) proposed modified hyperparameters for RigL that may yield higher generalization performance; however, a detailed investigation of these hyperparameters for SRigL is left to future work.

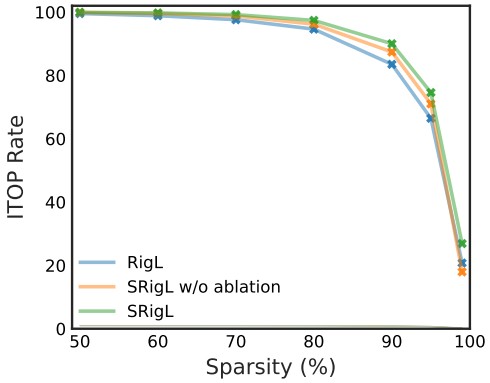

Figure 14: ResNet-18/CIFAR-10 ITOP rate

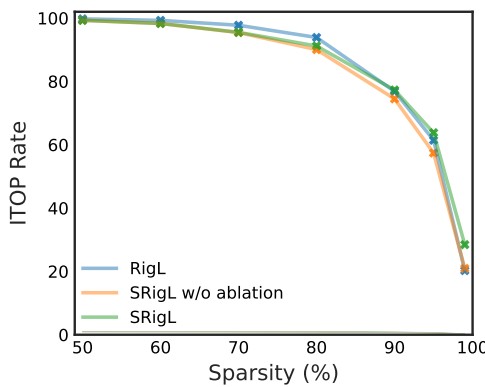

Figure 15: ResNet-18/CIFAR-10 ITOP rate

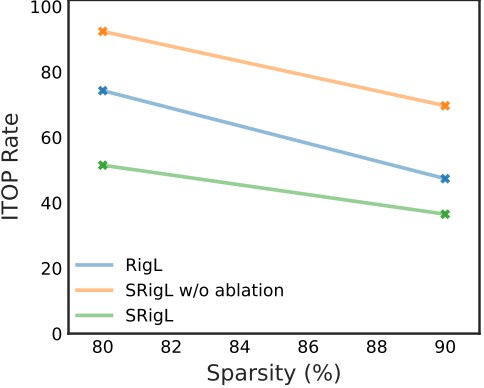

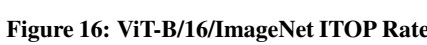

Figure 16: ViT-B/16/ImageNet ITOP Rate

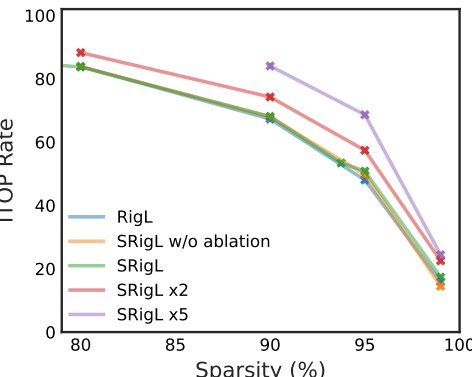

Figure 17: ResNet-50/ImageNet ITOP Rate

---

[2]MicroNet Challenge Github Repository

## I    CONDENSED LINEAR CPU BENCHMARK DETAILS

For each sparsity level, we used the trained weights from the last linear layer in the final multi-layer perception block from the ViT-B/16 transformer encoder. This layer has a width of 768 neurons and an input of 3072 features. The input and layer parameters are all set to a 32 bit floating point type. Across all sparsities, batch sizes, and number of threads investigated, our condensed representation utilizing both structured and fine-grained sparsity yields the fastest online inference speed. However, at higher batch sizes and modest sparsities, structured sparsity is often faster than our condensed representation. See Figs. 18 to 20 for benchmark results from 1-8 threads and batch sizes 1-64. We note that SRigL with either a condensed or a structured sparse representation yields the fastest benchmark times.

We used `torch.compile` with the inductor backend. For compiler options, we used the max-autotune mode and full graph output. However, full graph output is not compatible with CSR formats so we omit this parameter for the unstructured benchmarks. The benchmark script was run with a `niceness` value of $-15$ to ensure as accurate results as possible. The apparent slow down in 99% structured sparse benchmarks compared to other sparsities is due to the fact that SRigL ablates fewer neurons at 99% sparsity. At extreme sparsities, each neuron has very few active weights resulting in more neurons being considered as *salient* by SRigL.

## J    GPU BENCHMARKS

Using GPU CUDA kernels developed by (Schultheis & Babbar, 2023), we accelerate our sparse networks and demonstrate a significant acceleration for batched inference and a modest acceleration for online inference at high sparsities (>90%), see Fig. 21. All runs conducted on an NVIDIA Titan V. Note y-axis scale is logarithmic.

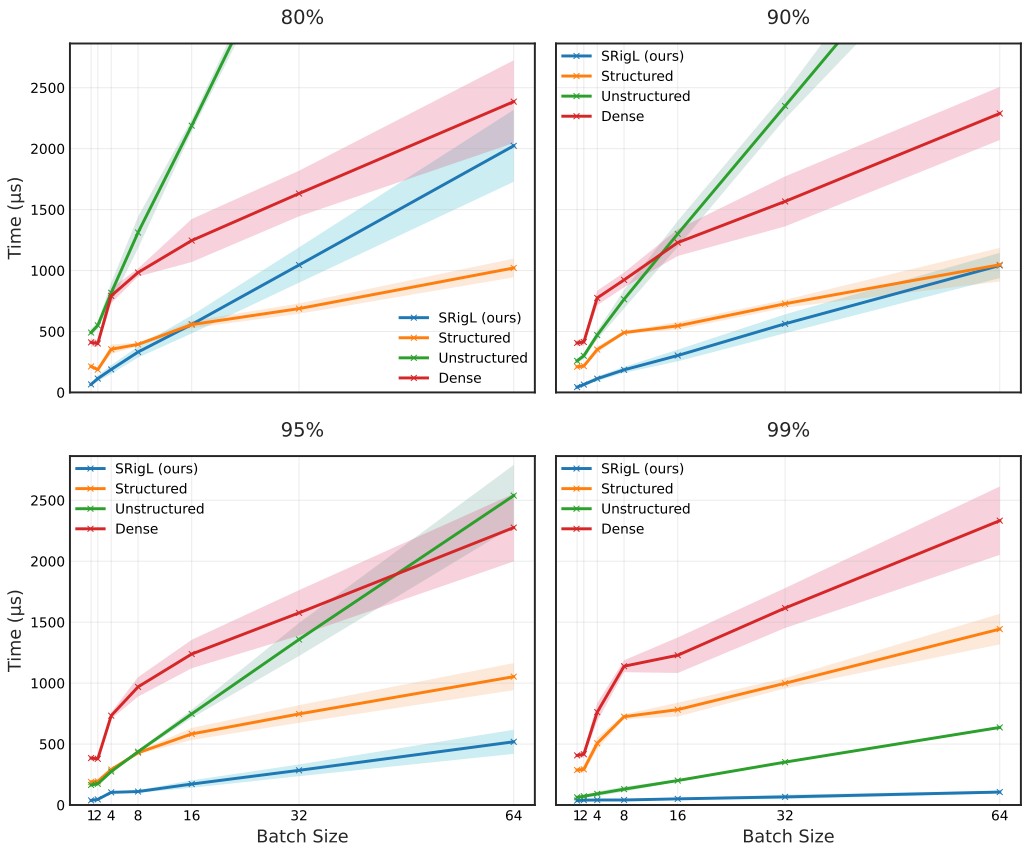

**Figure 18:** CPU benchmarks with 1 thread up to batch size 64

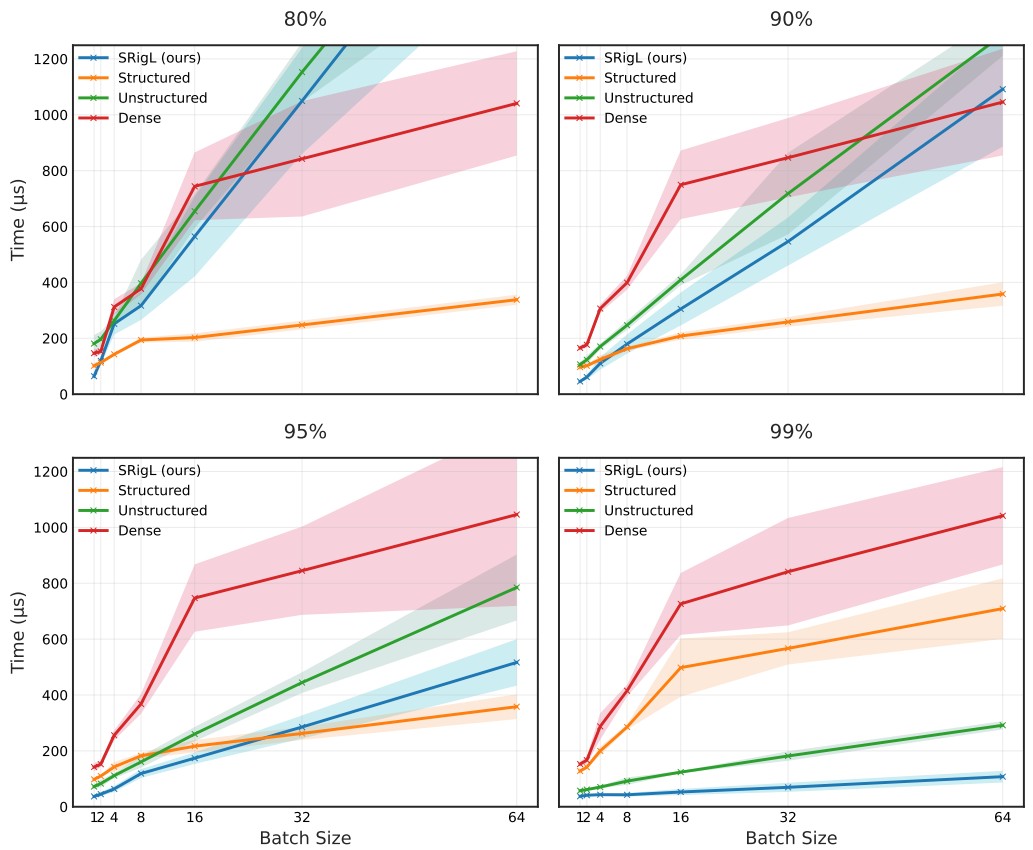

**Figure 19:** CPU benchmarks with 4 threads up to batch size 64

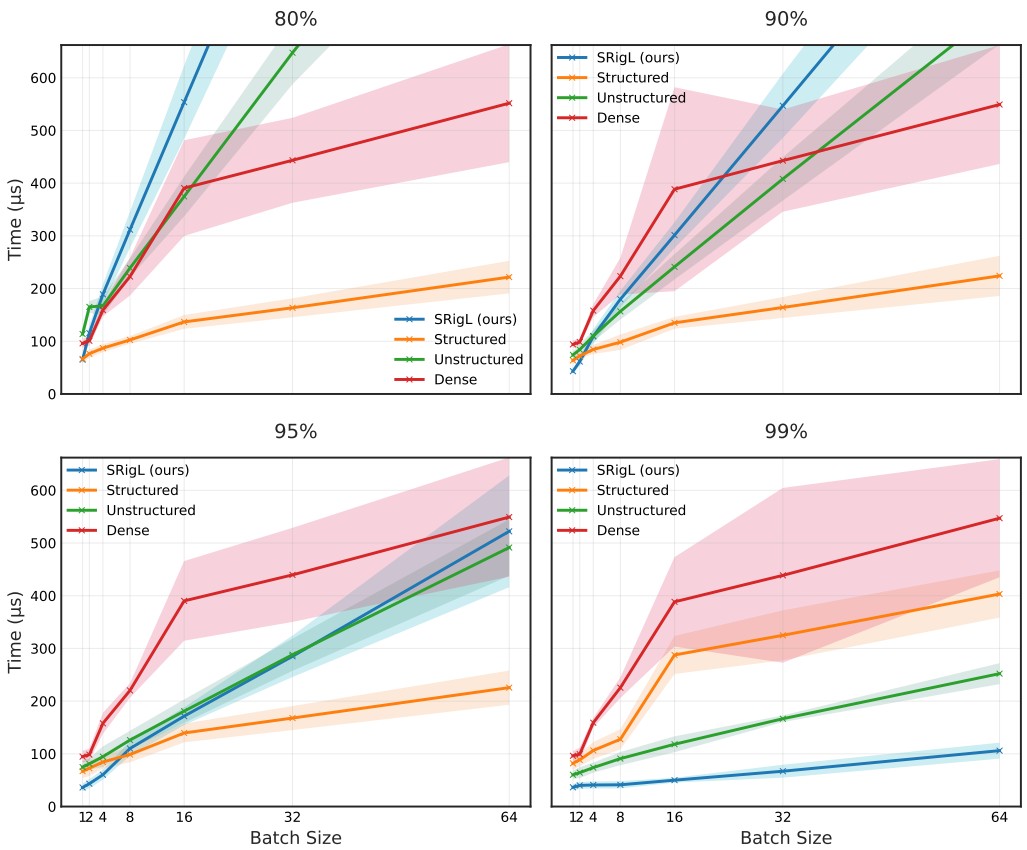

**Figure 20:** CPU benchmarks with 8 threads up to batch size 64

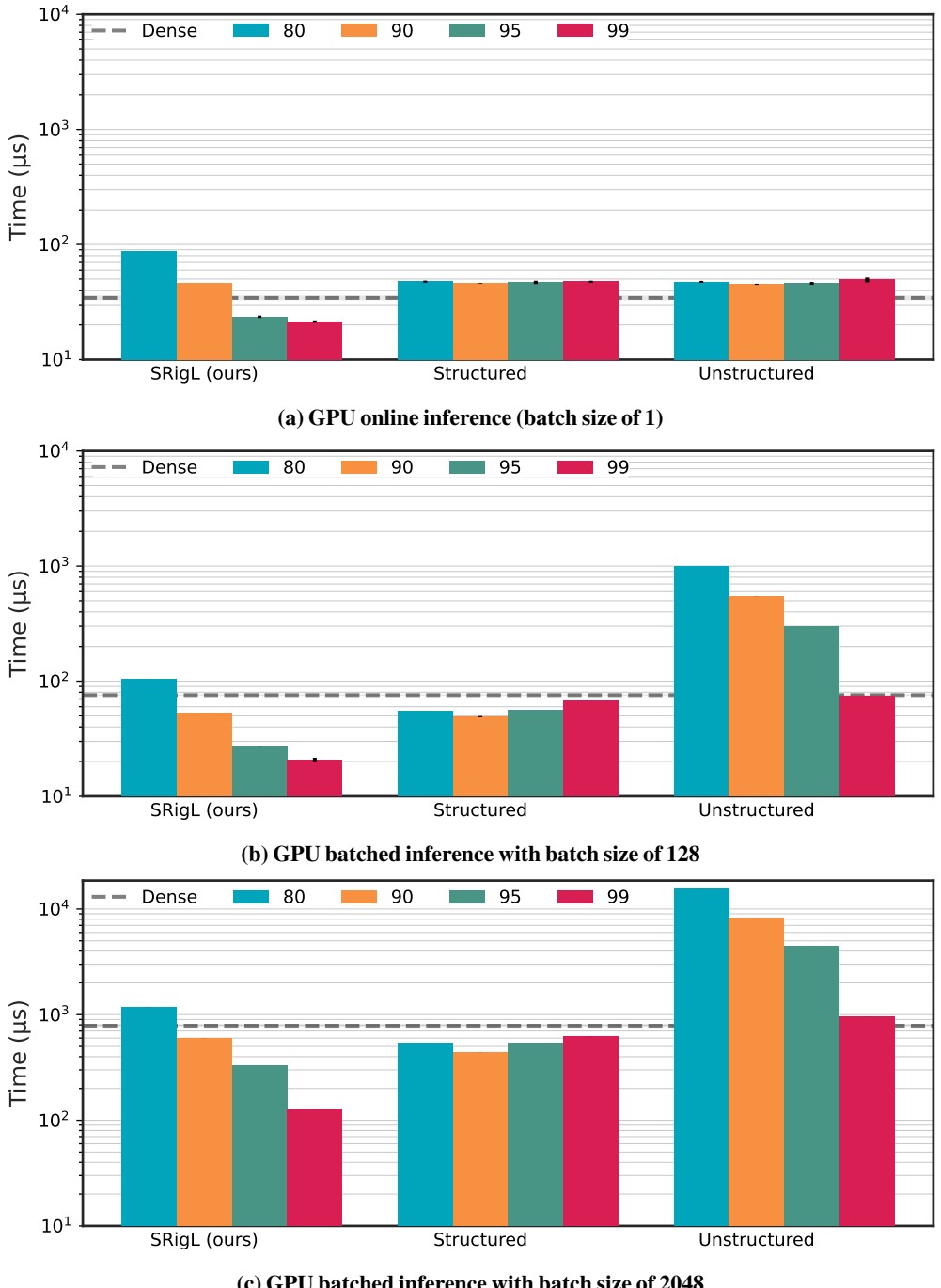

**(a) GPU online inference (batch size of 1)**

**(b) GPU batched inference with batch size of 128**

**(c) GPU batched inference with batch size of 2048**

**Figure 21: Real-world GPU wall-clock timings for inference** on an NVIDIA Titan V. We compare timings for a fully-connected layer extracted from the ViT-B/16 model trained with SRigL when compressed using the condensed representation learned by SRigL, structured (i.e. SRigL with only neuron ablation) and unstructured (i.e. CSR) representations. Batch sizes are 1, 256, and 2048 for sub-figures 21a, 21b, 21c, respectively. The median over a minimum of 5 runs is shown, while the error bars show the std. dev. Note: y-axis scale is logarithmic

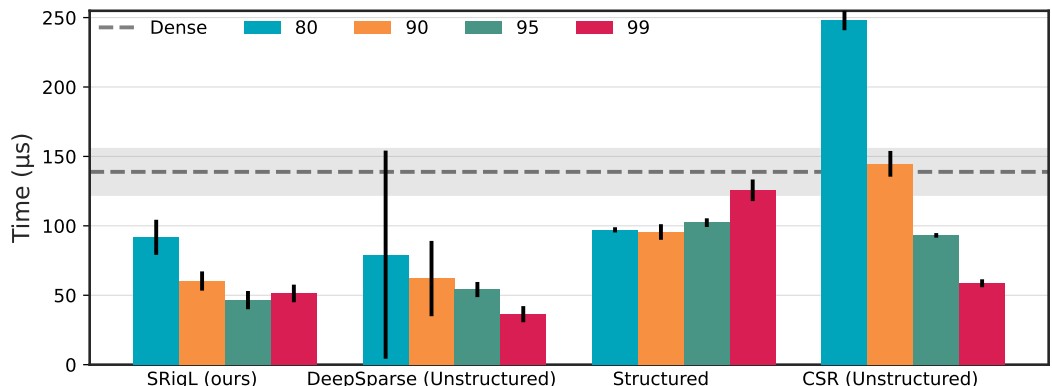

**Figure 22: Online inference with DeepSparse compared to SRigL** on an Intel Xeon W-2145 with 4 threads. The median over a minimum of 5 runs is shown, while the error bars show the std. dev.

## K DEEPSPARSE CPU BENCHMARKS

Here we present online inference benchmarks for CPU using the DeepSparse Engine library (Iofinova et al., 2021). DeepSparse library includes several engineering innovations to accelerate unstructured sparsity on CPU. For instance, a *depth-wise asynchronous* execution algorithm is used that takes advantage of the relatively large cache size for CPUs compared to hardware accelerators such as GPUs. Other additional innovations used include pre-loading the input data to hide latency via CPU *pipelining*, compressing sparse activations into a CSR format on-the-fly, and keeping convolutional kernels in L2 cache. For more details see Kurtz et al. (2020).

We compare our CPU timings for SRigL to DeepSparse in Fig. 22 and find similar latency; however, we note that DeepSparse is subject to a higher variability as evidenced by a larger standard deviation. Further, many of the innovations used to accelerate unstructured sparse networks with DeepSparse could equally be applied to networks trained with SRigL.

## L COMPARISON WITH STRUCTURED PRUNING METHODS

In the following table we compare several structured pruning methods to SRigL. The tabulated structured pruning methods typically prune and fine-tune a pretrained model, resulting in extended training duration compared to typical dense training. We report the inference FLOPs, top-1 accuracy, and number of epochs for each method in Table 10.

Table 10: **Top-1 ImageNet test accuracy of ResNet-50 for various structured pruning methods** compared with SRigL and Chase (Yin et al., 2023). All values, except for SRigL, are obtained from Yin et al. (2023).

| Methods | Inference FLOPs | Top-1 Accuracy | Epochs |
|---|---|---|---|
| Uniform | 2.0G | 75.1% | 300 |
| Random | 2.0G | 74.6% | 300 |
| GBN (You et al., 2019) | 2.4G | 76.2% | 350 |
| LEGR (Chin et al., 2020) | 2.4G | 75.7% | - |
| FPGM (He et al., 2019) | 2.4G | 75.6% | 200 |
| TAS (Dong & Yang, 2019) | 2.3G | 76.2% | 240 |
| Hrank (Lin et al., 2020) | 2.3G | 75.0% | 570 |
| SCOP (Tang et al., 2020) | 2.2G | 76.0% | 230 |
| CHIP (Sui et al., 2021) | 2.2G | 76.3% | - |
| Group Fisher (Liu et al., 2021a) | 2.0G | 76.4% | - |
| AutoSlim (Yu & Huang, 2019) | 2.0G | 75.6% | - |
| CafeNet-R (Su et al., 2021) | 2.0G | 76.5% | 300 |
| Chase-1[†](Yin et al., 2023) | 1.5G | 76.6% | 250 |
| SRigL[†] | 2.0G | 74.7% | 205 |
| SRigL[†] | 2.0G | 76.2% | 515 |
| Uniform | 1.0G | 73.1% | 300 |
| Random | 1.0G | 72.2% | 300 |
| Group Fisher (Liu et al., 2021a) | 1.0G | 73.9% | - |
| CafeNet-R (Su et al., 2021) | 1.0G | 74.9% | 300 |
| CafeNet-E (Su et al., 2021) | 1.0G | 75.3% | 300 |
| Chase-2[†](Yin et al., 2023) | 0.9G | 75.7% | 250 |
| SRigL[†] | 1.0G | 71.5% | 205 |
| SRigL[†] | 1.0G | 73.6% | 515 |

[†] DST methods. All other methods tabulated are structured pruning methods.

