# OpenReview forum: "Dynamic Sparse Training with Structured Sparsity"
_ICLR.cc/2024/Conference — ICLR 2024 poster_

### Official Review · Reviewer_HJFn · 2023-10-26

**Soundness:** 3 good
**Presentation:** 3 good
**Contribution:** 2 fair
**Rating:** 6
**Confidence:** 4

**Summary:**

This work proposes a dynamic sparse training method with constant fan-in constraint. The method is validated on several computer vision datasets and architectures.

**Strengths:**

The introduced sparsity pattern is quite flexible and allows achieving performance close to unstructured sparsity while being more hardware-friendly. Neuron ablation procedure looks very sound and improves noticeably performance of the method. The appendix involves theoretical analysis motivating the good optimization behavior of the proposed SRigL and the choice of constant fan-in sparsity pattern.

Method is validated on several models and attains pretty good performance at high sparsity.

The algorithm is simple and leads to speedups even without the need to write highly-customized code.

**Weaknesses:**

I think the proposed SRigL with constant fan-in sparsity should be compared to a similar sparse training procedure with N:M sparsity. One could perform it as follows: with some interval, a fraction of weights with the smallest magnitude among the group of M consecutive weights is dropped and the same fraction with high gradient magnitude is regrown. Indeed, the difference from original RigL is that the space of possible updates is much more constrained - i.e. for 2:4 sparsity one can prune a single non-zero weight inside the group of 4 and regrow one among the zeros. Likely, such strategy would perform poorly, but this comparison would motivate the necessity for constant fan-in sparsity. The other alternative is RigL with blocked sparsity that prunes whole groups of consecutive `block_size` weights. This sparsity pattern is known to be more CPU-speedup friendly compared to unstructured sparsity.

Method lacks comparison with dedicated inference engines that leverage sparsity. DeepSparse engine [1] achieves significant speed-ups for unstructured sparsity, especially on Intel CPUs.

*Minor* In the Appendix E plots w/o ablation seem to be absent on Figure 7.

---
[1] https://github.com/neuralmagic/deepsparse

**Questions:**

Wall-clock timings for structured sparsity look suspicious. Does it mean that N:M sparsity with higher sparsity may be slower than the one with lower sparsity? Indeed, it is hardly possible to achieve linear speed-up, but I would expect at most saturation for high N:M sparsity.

How well does the method perform when combined with quantization? One is expected to achieve even higher speed-ups for sparse+8bit quantized model.

How much does the method explore new connections compared to RigL? Would be interesting to compare some measure similar to In-Time Over-Parameterization, introduced in [1] for RigL and SRigL.

---
[1] https://arxiv.org/abs/2102.02887

---

> ### Author Response · Authors · 2023-11-17
> **Rebuttal to HJFn (1/2)**
>
> Thank you for your review.
>
> > I think the proposed SRigL with constant fan-in sparsity should be compared to a similar sparse training procedure with N:M sparsity. One could perform it as follows: with some interval, a fraction of weights with the smallest magnitude among the group of M consecutive weights is dropped and the same fraction with high gradient magnitude is regrown. Indeed, the difference from original RigL is that the space of possible updates is much more constrained - i.e. for 2:4 sparsity one can prune a single non-zero weight inside the group of 4 and regrow one among the zeros. Likely, such strategy would perform poorly, but this comparison would motivate the necessity for constant fan-in sparsity. The other alternative is RigL with blocked sparsity that prunes whole groups of consecutive block_size weights. This sparsity pattern is known to be more CPU-speedup friendly compared to unstructured sparsity.
>
> It has been previously demonstrated that N:M sparsity can be learned without significant generalization performance degradation in the dense-to-sparse DST paradigm. Further, you are correct that as M decreases, model performance tends to degrade. At the limit, unstructured sparsity takes M to equal an entire layer of weights. We fall somewhat in between, with constant fan-in effectively being an N:M sparsity pattern where M is the dense fan-in and N is the sparse fan-in.
>
> While we agree that this would be an interesting experimental setup, we have motivated why we believe constant fan-in is worth exploring in Section 4, *even if N:M sparsity can be learned in a DST setting*.  To briefly summarize the noted benefits of constant fan-in vs. N:M sparsity:
> * Flexibility of constant fan-in to arbitrary sparsity ratios
> * Enabling the use of layer-wise sparsity distributions rather than a single global sparsity level
> * Hardware support for N:M sparsity is strictly limited to 2:4 sparsity at this time. However, we demonstrate speed-ups for arbitrary sparsity ratios *without* the use of any specialized hardware.
>
> In any case, we hope to explore a wider variety of fine-grained and structural constraints in our upcoming work intended to compliment this paper.
>
> > Method lacks comparison with dedicated inference engines that leverage sparsity. DeepSparse engine [1] achieves significant speed-ups for unstructured sparsity, especially on Intel CPUs.
>
> Time permitting, we will add a comparison to DeepSparse to our updated paper prior to the end of the rebuttal period.
>
> > Minor In the Appendix E plots w/o ablation seem to be absent on Figure 7.
>
> Thank you, we will update the plot in our revised paper.
>
> > Wall-clock timings for structured sparsity look suspicious. Does it mean that N:M sparsity with higher sparsity may be slower than the one with lower sparsity? Indeed, it is hardly possible to achieve linear speed-up, but I would expect at most saturation for high N:M sparsity.
>
> We believe you are referring to Figure 4, which shows increased wall-clock timings for structured sparsity at 99% vs. 80-95%. The key consideration here is that the sparsity level refers to the overall sparsity of the network, *not the structured sparsity level*. Specifically, for the 99% sparse model, there are 482 neurons active whereas there are 344, 318, and 322 neurons active at sparsity levels 95, 90, and 80, respectively. SRigL does not enforce any specific sparsity requirements on the structural constraint, rather the structural sparsity is learned based on gamma-sal. For highly sparse models such as the 99% case here, we have found that this methodology will yield relatively fewer ablated neurons since most neurons have very few sparse weights and thus a higher percentage of those weights are likely to be considered salient, prohibiting ablation of the corresponding neuron.
>
> This is stated in Appendix H, albeit briefly. Here’s the relevant excerpt:
> “The apparent slow down in 99% structured sparse benchmarks compared to other sparsities is due to the fact that SRigL ablates fewer neurons at 99% sparsity. At extreme sparsities, each  neuron has very few active weights resulting in more neurons being considered as salient by SRigL.”
>
> > How well does the method perform when combined with quantization? One is expected to achieve even higher speed-ups for sparse+8bit quantized model.
>
> Time permitting, we will include an 8-bit quantized model in our wall-clock benchmarks for the revised paper. We speculate at this stage that 8-bit quantization will decrease wall-clock timings for all compressed representations examined (SRigL, structured, CSR).

---

> ### Author Response · Authors · 2023-11-17
> **Rebuttal to HJFn (2/2)**
>
> > How much does the method explore new connections compared to RigL? Would be interesting to compare some measure similar to In-Time Over-Parameterization, introduced in [1] for RigL and SRigL.
>
> We measured ITOP for all networks during training. We agree that the ITOP rate is of interest and we will include an additional appendix section with these values. To summarize at this stage for the purposes of discussion:
> * For ResNet 18 and 50, we find that SRigL and RigL have very comparable ITOP rates. Overall the final ITOP value is primarily predicated on the sparsity level. Ie.., at 80% sparsity Rigl/SRigl achieve a final ITOP rate of about 0.8 and almost 1.0 at 50% sparsity.
> * For ViT, SRigL has a significantly lower ITOP rate of about 0.4 and 0.5 at 90% and 80% sparsity, respectively. In comparison, RigL achieves an ITOP rate of 0.48 and 0.74 at the same corresponding sparsity levels. We see much higher ITOP rates if ablation is disabled, suggesting that much of the parameter space that remains unexplored is present in ablated neurons.
> * For MobileNet, initial experiments suggest a 0.1 to 0.2 increase in ITOP for SRigL vs. RigL.

---

> > ### Comment · Reviewer_HJFn · 2023-11-18
> >
> > Thanks for the clarifications and comments. I read your response and had a look at the revised version of the manuscript. Since my concerns were addressed, I decide to raise the score.

---

### Official Review · Reviewer_qWXP · 2023-10-30

**Soundness:** 3 good
**Presentation:** 3 good
**Contribution:** 2 fair
**Rating:** 6
**Confidence:** 2

**Summary:**

The paper introduces Structured RigL (SRigL), an advancement in Dynamic Sparse Training (DST). SRigL combines structured sparsity with neural network training, resulting in models that are both computationally efficient and performant. The method achieves faster real-world inference times on CPUs and outperforms other sparse training techniques in various neural network architectures.

**Strengths:**

(1) SRigL creatively combines structured N:M sparsity with a constant fan-in constraint which is from existing DST methodologies.

(2) The paper provides extensive empirical evidence to show its superior performance.

(3) SRigL's ability to achieve faster real-world inference times on CPUs is of paramount importance.

**Weaknesses:**

(1) an important baseline is missing [1].

(2) Lack of novelty: A combination of sparse training and N: M sparsity have been shown in previous study [1].

[1] Yin, Lu, et al. "Dynamic Sparsity Is Channel-Level Sparsity Learner."  NeurIPS 2023.

**Questions:**

(1) with comparison to [1], what's the advantages of SRigL?

(2) Why AST and SR-STE is not comparable to SRigl?  it would be better to report the results of AST and SR-STE since they are very related works.

---

> ### Author Response · Authors · 2023-11-17
> **Rebuttal to qWXP (1/2)**
>
> Thank you for your review.
>
> > (1) an important baseline is missing [1]...Yin, Lu, et al. "Dynamic Sparsity Is Channel-Level Sparsity Learner." NeurIPS 2023.
>
> Thank you for providing this reference. Indeed, Yin et al.’s observation of the existence of sparse-amenable channels matches our own. However, we explore a combination of structured with fine-grained sparsity in addition to using very different implementation approaches. For instance, Chase includes several important additional components to obtain their reported performance metrics as detailed in their ablation study (Fig. 4 of their paper). Most notably, they use a gradual pruning schedule (dense-to-sparse) and a soft memory bound. While these components improve generalization performance, they result in a higher memory footprint during training than truly end-to-end sparse training.
>
> Furthermore, we find that our work is complementary to Chase as we explore a different channel level saliency criterion (min. % saliency weights vs. UMM in Chase) and also demonstrate that pruning-amenable channels can be exploited **in addition to fine-grained sparsity**. This is an important finding in its own right as it suggests that we can obtain benefits of both structured and fine-grained sparsity, without incurring the cost of retraining or fine-tuning.
>
> Finally, it is worth noting that Chase is a *contemporaneous* work as it was published within the last four months. As per the ICLR 2024 FAQ for reviewers, we are not required to compare to Chase directly. In any case, we are happy to cite Chase in our camera-ready version to complement our work and provide further evidence for the emergence of structure during unstructured sparse training.
>
> > (2) Lack of novelty: A combination of sparse training and N: M sparsity have been shown in previous study [1].
>
> Chase does **not** use N:M sparsity as far as we are aware, so we are unable to respond to this question specifically. Could you clarify which study you are referring to?
>
> However, sparse training has been previously demonstrated with N:M sparsity in general. As we note in Section 4, there are some specific benefits of constant fan-in sparsity over N:M sparsity that warrant further consideration:
> * Flexibility of constant fan-in to arbitrary sparsity ratios
> * Enabling the use of layer-wise sparsity distributions rather than a single global sparsity level
> * Hardware support for N:M sparsity is strictly limited to 2:4 sparsity at this time. However, we demonstrate speed-ups for arbitrary sparsity ratios *without* the use of any specialized hardware.
>
> Further to the above, we are the first work to: investigate the constant fan-in constraint in detail, demonstrate learning a hybrid structured + fine-grained sparsity, and we provide further evidence regarding the existence of neuron ablation in unstructured DST methods. In our opinion, these contributions represent a reasonable degree of novelty to justify publication.
>
> > (1) with comparison to [1], what's the advantages of SRigL?
>
> We include both structured and fine-grained sparsity in our method, enabling further acceleration during inference. We also developed different saliency measures for determining when to ablate neurons / channels. Finally, our method does not require setting the target % channels to prune (Sc) as the number of channels pruned is actually learned directly by SRigL. See above for more details.
>
> In our opinion, our work compliments Chase by: 1) providing further evidence of neuron ablation in unstructured DST methods; 2) introducing the constant fan-in constraint; 3) demonstrating learning a hybrid structured + fine-grained sparsity.

---

> ### Author Response · Authors · 2023-11-17
> **Rebuttal to qWXP (2/2)**
>
> > (2) Why AST and SR-STE is not comparable to SRigl? it would be better to report the results of AST and SR-STE since they are very related works.
>
> Algorithms such as SET, RigL, and SRigL are sparse-to-sparse DST algorithms, whereas SR-STE and AST are dense-to-sparse algorithms. There are some important advantages of sparse-to-sparse training compared to dense-to-sparse training:
>
> * Sparse-to-sparse training has lower computational overhead as the network is trained with the sparse weights throughout training. The advantages of sparsity can be realized immediately during the training process rather than only being reserved for inference.
> * Sparse-to-sparse training has lower memory overhead since the non-active weights can be removed before training begins. This enables training of sparse models for which a dense counterpart would not be possible to represent at a fixed memory budget. As we have seen in recent years, increasing scale yields significant benefits. Sparse-to-sparse algorithms may be able to push model scale even further than the current limits.
> * SR-STE also updates the network connectivity every mini-batch as opposed to every 100 mini-batches. This adds additional computational overhead during training and requires storing the dense parameters throughout training.
>
> We hope we have made a clear case that sparse-to-sparse algorithms have some important properties that cannot be replicated with a dense-to-sparse algorithm. As network training costs continue to grow, we believe the advantages of sparse-to-sparse training are worth continuing to explore despite recent successes in the dense-to-sparse paradigm.
>
> In terms of reporting these results, we have endeavored to do so where possible. SR-STE results are included in Table 3 and we will add AST as well for clarity. We note that SRigL achieves results comparable to AST on Resnet50/Imagenet at 80/90% sparsity. See Table 3 of AST paper for this comparison.

---

> ### Comment · Reviewer_qWXP · 2023-11-22
>
> Thank you for your response. Your response has addressed my concerns and I will keep my score.

---

### Official Review · Reviewer_UhkR · 2023-10-31

**Soundness:** 3 good
**Presentation:** 3 good
**Contribution:** 2 fair
**Rating:** 6
**Confidence:** 3

**Summary:**

The paper proposes a method for dynamic sparse training that leads to structured N:M sparsity, here realized as a constant fan-in degree for neurons. The authors modify RigL to achieve this. The proposed constant fan-in N:M sparsity can theoretically achieve faster inference speeds on new GPU chips that are produced for general consumers but can specifically accelerate this type of structure.

The authors show that the output norm variance can in expectation be reduced with constant fan-in sparsity. The authors experimentally show that their proposed SRigL leads to the similar performance as RigL while enforcing the imposed structure.

**Strengths:**

The paper tackles an important issue in deep learning, is well-motivated, and written in a clear fashion. The motivation for the benefits of structured sparsity are clear. The main method is simple to explain, and seems to work approximately as good as RigL.
I have reviewed the paper before, and appreciate that the authors include a wall-clock time comparison of a fully connected layer on a CPU.

**Weaknesses:**

I have reviewed this paper before and still remain unconvinced by the author's argumentation on the main benefits of SRigL. While the authors claim that sparse-to-sparse training such as SRigL is beneficial over dense-to-sparse methods in terms of memory usage and computational time, the evidence presented for this is marginal. While the benefit of sparse models at inference time is obvious, the benefit of sparsifying models at training time (that reach the same generalization error) should be either faster training times or lower memory footprint. The authors show no evidence that SRigL trains models faster than dense-to-sparse methods such as SR-STE. On a similar note, a real-world quantitative evaluation of the difference in memory footprint for dense-to-sparse methods (such as SR-STE) vs. SRigL would make this paper much more convincing.

**Questions:**

For the wall-clock time comparison, it seems that the median for at least 5 runs is shown. Given that the inference time is in the microsecond range, why not show the median of millions of forward passes, to reduce noise?

What should be the main takeaway from you showing SRigL x2 and SRigL x5 in Figure 3? It seems to be included to inflate the presentation of results of SRigL, but RigL performs the same under x2 or x5 training times. Either include RigL x2 and RigL x5, or omit both from Figure 3.

**Details Of Ethics Concerns:**

-

---

> ### Author Response · Authors · 2023-11-17
> **Rebuttal to UhkR**
>
> Thank you for your second review!
>
> > I have reviewed this paper before and still remain unconvinced by the author's argumentation on the main benefits of SRigL. While the authors claim that sparse-to-sparse training such as SRigL is beneficial over dense-to-sparse methods in terms of memory usage and computational time, the evidence presented for this is marginal. While the benefit of sparse models at inference time is obvious, the benefit of sparsifying models at training time (that reach the same generalization error) should be either faster training times or lower memory footprint. The authors show no evidence that SRigL trains models faster than dense-to-sparse methods such as SR-STE. On a similar note, a real-world quantitative evaluation of the difference in memory footprint for dense-to-sparse methods (such as SR-STE) vs. SRigL would make this paper much more convincing.
>
> Unfortunately, the engineering effort to realize an end-to-end acceleration of SRigL is out of the scope of this work. We have endeavored to demonstrate that, **in theory**, sparse-to-sparse methods such as SRigL can be accelerated and compressed during training. This is a tantalizing possibility as it will enable larger-sparse models to be trained that would otherwise not be possible on a fixed VRAM budget. We agree that showing this acceleration in practice is ideal; however, our goal with this work is to establish whether the proposed constant fan-in constraint negatively affects generalization performance. Having established this in this work, we are now working on demonstrating real-world end-to-end training speed-ups in an upcoming, complimentary work.
>
> As an aside, the SR-STE repository has not been updated in over a year and we have had trouble getting the authors to clarify their build process. Notably, they did not include a requirements file or similar implementation details as required to reproduce their results (ie., they have not published code for language models, object detection, nor instance segmentation).
>
> Furthemore, while the SR-STE authors published theoretical FLOPs calculation, they did not include any runtime characteristics which may have been used for a direct comparison.
>
> > For the wall-clock time comparison, it seems that the median for at least 5 runs is shown. Given that the inference time is in the microsecond range, why not show the median of millions of forward passes, to reduce noise?
>
> We used a blocked autorange with a minimum number of runs set to 5 for our benchmarks. For the results reported, the median number of runs was 438 and the minimum was 15. Given the relatively narrow standard deviation ranges, we believe this is sufficient to demonstrate the statistical significance of our results.
>
> > What should be the main takeaway from you showing SRigL x2 and SRigL x5 in Figure 3? It seems to be included to inflate the presentation of results of SRigL, but RigL performs the same under x2 or x5 training times. Either include RigL x2 and RigL x5, or omit both from Figure 3.
>
> The main takeaway is that, like RigL and other DST methods, SRigL can be trained for longer to reach higher generalization performance; enabling the use of sparsities up to 90% while matching the generalization performance of a dense network. We also wanted to demonstrate how training the network for longer yields more neurons being ablated, similar to RigL’s off the shelf ablation. These are salient details for the reader that are not intended to show a benefit of SRigL over RigL but rather to demonstrate that the extended training behavior is similar to RigL. We note that we explicitly state that “SRigL yields **similar** generalization performance as RigL across each sparsity and training duration considered…**Similar to RigL**, we observe that SRigL generalization performance improves with increasing training time”. We hope that our intent is made clear by these statements.
>
> Due to compute limitations, we were unable to train RigL for extended durations on our own code base and therefore do not include it in our plots. We do however report RigL’s extended training duration results in Table 1.

---

### Official Review · Reviewer_LY1T · 2023-11-01

**Soundness:** 3 good
**Presentation:** 2 fair
**Contribution:** 3 good
**Rating:** 5
**Confidence:** 4

**Summary:**

This paper presents Structured RigL (SRigL), a Dynamic Sparse Training (DST) method that excels in training sparse neural networks. SRigL achieves state-of-the-art performance in DST by incorporating fine-grained N:M sparsity and continuous fan-in constraints for sparse interstructures. Through heuristic analysis and neuron removal, SRigL outperforms existing methods across various neural network architectures, demonstrating a substantial 3.6x/2x speedup on CPU at 90% sparsity compared to equivalent dense or unstructured sparse layers.

**Strengths:**

* The authors propose SRigL method that learns a SNN with constant fan-in fine-grained structured sparsity while maintaining generalization comparable to RigL even at high sparsity levels across various network architectures.
* Experimental results show that the SRigL method can not only improve the efficiency of parameters and memory, but can also enable acceleration during training.
* The proposed SRigL demonstrates minimal accuracy drop even at high sparsity levels exceeding 90% in both ResNet and ViT architectures.

**Weaknesses:**

* The proposed SRigL primarily demonstrates its efficacy on networks with relatively high redundancy, such as ResNet or ViT. However, there is a lack of experimentation on networks with lower redundancy, such as MobileNet.
* The experimental results are limited to vision tasks.
* While the comparison from the perspective of Dynamic Sparse Training (DST) in Table 3 is crucial, I believe that the results regarding structured pruning performance are equally significant. However, there is a notable absence of experiments, and comparisons with other techniques with 'structured' attributes are challenging.
* A minor concern is the absence of a comprehensive figure that provides an overview of the entire process of SRigL.

**Questions:**

* In Section 4.4, the description of Algorithm 1 mentions, "The algorithm to accelerate our condensed sparsity representation is shown in Algorithm 1, demonstrating its embarrassingly parallel nature." I'm interested in knowing the throughput on real GPU or CPU based on the sparsity levels in the matmul unit-test.
* While discussing "Constant fan-in sparsity," it occurs to me that there might be performance variations across tasks and networks depending on the information included by input features. Have there been any experiments applying this approach to tasks other than vision?
* It is anticipated that the proposed SRigL method is significantly influenced by the $\gamma_{sal}$ value. I think this sensitivity might be more pronounced in networks with lower redundancy, such as MobileNet. Has there been an observation of trends by sweeping through different $\gamma_{sal}$ values? If so, what were the findings?
* How does the application of SRigL affect the practical outcomes (latency, throughput) in the context of "end-to-end sparse training" on GPUs?

---

> ### Author Response · Authors · 2023-11-17
> **Rebuttal to LY1T (1/2)**
>
> Thank you for your detailed review and suggestions to improve the paper.
>
> > The proposed SRigL primarily demonstrates its efficacy on networks with relatively high redundancy, such as ResNet or ViT. However, there is a lack of experimentation on networks with lower redundancy, such as MobileNet.
>
> Thank you for this suggestion. We are currently running MobileNet experiments and can confirm that initial results show RigL and SRigL have comparable performance. We will update our paper with the mobilenet results as soon as they are completed.
>
> > The experimental results are limited to vision tasks.
>
> The goal of this paper is to introduce the constant fan-in constraint and our observations and methods related to neuron ablation. We agree that a more diverse task set is preferable; however, in our opinion the empirical evidence provided is reasonably compelling for inclusion in the conference. Future work will aim to exploit similar methods to SRigL on other tasks.
>
> > While the comparison from the perspective of Dynamic Sparse Training (DST) in Table 3 is crucial, I believe that the results regarding structured pruning performance are equally significant. However, there is a notable absence of experiments, and comparisons with other techniques with 'structured' attributes are challenging.
>
> Our primary intent was to investigate constant fan-in in the context of Dynamic Sparse Training. However, we agree this is an interesting perspective and will add a table summarizing current SoTA structural pruning methods vs. SRigL to the revised paper before the end of the discussion period.
>
> > A minor concern is the absence of a comprehensive figure that provides an overview of the entire process of SRigL
>
> We were somewhat limited by the page limit for this initial submission but will include an overview figure in the revised figured to help clarify the high-level details of our method.
>
> > In Section 4.4, the description of Algorithm 1 mentions, "The algorithm to accelerate our condensed sparsity representation is shown in Algorithm 1, demonstrating its embarrassingly parallel nature." I'm interested in knowing the throughput on real GPU or CPU based on the sparsity levels in the matmul unit-test.
>
> To be clear, our results in Figure 4 and appendix H are on a real-world CPU. In addition, we will include GPU benchmarks in our revised paper before the end of the rebuttal deadline. We can report that our GPU implementations significantly outperform both the dense and CSR benchmarks, especially at high batch sizes and high sparsities.
>
>  We are also working on benchmarking a Raspberry Pi 4 as a prototypical example of an edge device. We hope to include these experimental results in our revised rebuttal paper before the end of the discussion period.
>
> > While discussing "Constant fan-in sparsity," it occurs to me that there might be performance variations across tasks and networks depending on the information included by input features. Have there been any experiments applying this approach to tasks other than vision?
>
> Not at this time. Our initial goal was to establish whether the proposed constant fan-in constraint would significantly affect generalization performance on a variety of network architectures. We intend to explore more diverse tasks in an upcoming complimentary work.
>
> > It is anticipated that the proposed SRigL method is significantly influenced by the gamma-sal value. I think this sensitivity might be more pronounced in networks with lower redundancy, such as MobileNet. Has there been an observation of trends by sweeping through different gamma-sal values? If so, what were the findings?
>
> In our experiments we found that gamma-sal did not significantly affect generalization performance. The critical threshold was to enable ablation by requiring *a single salient weight per neuron*. However, we found that this integer limit did not transfer well between network architectures. For instance, for convolutional networks the sparse fan-in per neuron is on the order of 5-20. But for ViT the sparse-fan in is 1 to 2 orders of magnitude larger than our convolutional network experiments. Therefore, the gamma-sal hyperparameter enables us to set a threshold that takes the network topology into consideration.
>
> We did perform sweeps on the gamma-sal value. See Appendix E, Figures 6.a), 6.b), and 7. You will note the low variance in performance w.r.t. gamma-sal for the ResNet networks. ViT has a higher variance of about 3% in final performance when higher gamma-sal values are used.

---

> ### Author Response · Authors · 2023-11-17
> **Rebuttal to LY1T (2/2)**
>
> > How does the application of SRigL affect the practical outcomes (latency, throughput) in the context of "end-to-end sparse training" on GPUs?
>
> SRigL can be used to improve latency / throughput during training by exploiting the structured sparsity that is included in the method. The forward / backward passes can be accelerated for 99% of mini-batches, whenever a topology update is not required. Every $\Delta$T steps the network would need to uses the dense gradients for calculating the regrown scores. However, it is trivial to extend SRiGL to SET or other DST methods which do not require dense gradient information; in this case, every forward and backward pass can be accelerated while taking advantage of the hybrid structured + fine-grained we introduce in this work.
>
> Additionally, the constant fan-in constraint can be used at inference time in addition to realize further decreases in latency.

---

> ### Comment · Reviewer_LY1T · 2023-11-22
>
> Thank you for the detailed answers and results.
>
> I believe that the proposed SRigL method can achieve acceleration not only during inference but also in the training phase by reducing computation costs. If the proposed method had been compared to RigL in terms of throughput improvement during DST execution on a GPU, I would have gladly raised my score. However, I think that I did not receive sufficient answers to this question in the author's response, so I would like to keep my rating.

---

> > ### Author Response · Authors · 2023-11-22
> >
> > We compare SRigL to RigL in our timing benchmarks for *inference* in Figures 4, 12-15, and 20. In these figures the "unstructured" series represent RigL in a condensed form using CSR or the DeepSparse Engine. We can see that the structured + fine-grained sparsity learned by SRigL is much faster than the unstructured sparsity learned by RigL. These results closely approximate acceleration of the forward pass during training.
> >
> > During training, SRigL is more amenable to acceleration than RigL in theory due to the combination of structured and fine-grained sparsity. However, we do not currently exploit a condensed representation during training; therefore, RigL and SRigL have similar runtime characteristics during training as presently implemented. For the purposes of this work, our primary goal was to establish that the constant fan-in constraint did not significantly impair generalization performance and that such a sparsity constraint could yield acceleration for inference. We believe that these primary contributions are deserving of acceptance in their own right.
> >
> > In our upcoming follow-up work, we are working to demonstrate wall-clock training acceleration for SRigL compared to other DST methods and dense benchmarks. Unfortunately, the non-trivial engineering effort required to achieve this acceleration is considered out of scope for this work.

---

### Official Review · Reviewer_43nv · 2023-11-01

**Soundness:** 3 good
**Presentation:** 3 good
**Contribution:** 3 good
**Rating:** 8
**Confidence:** 4

**Summary:**

This paper proposes SRigL with a type of structured sparsity - constant fan-in sparsity that can be applied to dynamic sparse training, leading to real-world clock-time savings while maintaining the optimal performance.

**Strengths:**

1. The research topic of structured DST is timely and important for the ML community. With the popularity of LTH and DST, sparse neural networks have received upsurging attention due to its promising capacity to reduce training/inference costs while maintaining the original performance of dense counterparts. However, the benefits of sparse NNs can largely constraint by the limited support from common hardware - GPUs. Research works to improve the progress of this direction make significant contributions to the community.

2. This paper provides a comprehensive and precise related work that covers the most state-of-the-art structured/unstructured sparse training approaches. I very much appreciate such rich related works that provide enough credits and credits to previous works.

3. The detailed step of SRigL in Section 3 provides a good overview to understand the methodology.

4. SRigL is able to find small-sparse NNs that enjoy better real-world wall-clock for online inference than structured (SRigL with only neuron ablation) and unstructured NNs.

5. Compared with N:M sparsity, SRigL enjoys uniform layer-wise sparsity, which is more desirable for performance.

**Weaknesses:**

(1) Partial of the ideas used in SRigL has some certain levels of overlaps with the previous work (Chase: https://arxiv.org/pdf/2305.19454.pdf). For instance, Chase also uncovers that a large proportion of channels (termed sparse amenable channels) tend to be sparse during DST. They also perform channel pruning to produce a mixture of structured and unstructured sparsity at the end of training. It is better to clarify the difference and similarity between Chase and SRigL, even though Chase does not introduce the hardware-friendly constant fan-in sparsity for the unstructured part.

(2) Can SRigL also accelerate the training process for the online inference, with real-world wall-clock saving?

(3)  In many cases in Table 1 and 3, SRigL w/ ablation even outperforms SRigL w/o ablation, which is a bit counter-intuitive. Cause SRigL w/ ablation essentially produces a smaller-sparse model if I understand correctly, which would decrease the model capacity. Can the authors elaborate more about this? Does  SRigL w/o ablation means only pruning these dean channels without weight regrowing?

**Questions:**

(1) What is the technical difference between SRigL and Chase?

(2) Can SRigL also accelerate the training process for the online inference, with real-world wall-clock saving? Moreover, I noticed that the Constant Fan-in sparsity can be accelerated by GPUs with some custom CUDA implementation, such as Schultheis & Babbar (2023). I am wondering how difficult to make SRigL accelerated by GPUs using the CUDA implementation provided by Schultheis & Babbar (2023).

(3) Why SRigL w/ ablation outperforms SRigL w/o ablation? I suppose that SRigL w/o ablation is essentially the unstructured version of RigL.

(4) Since the real-world clock-time is measured on CPU. I am wondering how different it is to implement GPU kernel to support SRigL in common GPUs?

Overall, I think this paper is a good asset and I am willing to increase my score if the above weaknesses can be resolved.

---

> ### Author Response · Authors · 2023-11-17
> **Rebuttal to 43nv (1/2)**
>
> Thank you for your detailed review and insightful questions.
>
> > (1) Partial of the ideas used in SRigL has some certain levels of overlaps with the previous work (Chase: https://arxiv.org/pdf/2305.19454.pdf). For instance, Chase also uncovers that a large proportion of channels (termed sparse amenable channels) tend to be sparse during DST. They also perform channel pruning to produce a mixture of structured and unstructured sparsity at the end of training. It is better to clarify the difference and similarity between Chase and SRigL, even though Chase does not introduce the hardware-friendly constant fan-in sparsity for the unstructured part.
>
> Thank you for providing this reference. Indeed, Yin et al.’s observation of the existence of sparse-amenable channels matches our own. However, we explore a combination of structured with fine-grained sparsity in addition to using very different implementation approaches. For instance, Chase includes several important additional components to obtain their reported performance metrics as detailed in their ablation study (Fig. 4 of their paper). Most notably, they use a gradual pruning schedule (dense-to-sparse) and a soft memory bound. While these components improve generalization performance, they result in a higher memory footprint during training than truly end-to-end sparse training.
>
> Furthermore, we find that our work is complementary to Chase as we explore a different channel level saliency criterion (min. % saliency weights vs. UMM in Chase) and also demonstrate that pruning-amenable channels can be exploited **in addition to fine-grained sparsity**. This is an important finding in its own right as it suggests that we can obtain benefits of both structured and fine-grained sparsity, without incurring the cost of retraining or fine-tuning.
>
> Finally, it is worth noting that Chase is a *contemporaneous* work as it was published within the last four months. As per the ICLR 2024 FAQ for reviewers, we are not required to compare to Chase directly. In any case, we are happy to cite Chase in our camera-ready version to complement our work and provide further evidence for the emergence of structure during unstructured sparse training.
>
> > (2) Can SRigL also accelerate the training process for the online inference, with real-world wall-clock saving?
>
> In principle yes, we can accelerate the training process by compressing our model to a structured sparse representation for the forward and backward passe for all mini-batches between $\Delta$T. However, implementation of this acceleration has not yet been completed. We are currently exploring this in a separate work. Fine-grained sparsity in general is more challenging to accelerate on the backward pass; typically it is required that the fine-grained structured is transposable (ie.,sparsity  constraint applies equally to rows and columns of weight matrices). Since we do not not exploit a transposable constraint in this work, we cannot accelerate the fine-grained structure in practice with currently available commodity hardware. See [1] for more details.
>
> > (3) In many cases in Table 1 and 3, SRigL w/ ablation even outperforms SRigL w/o ablation, which is a bit counter-intuitive. Cause SRigL w/ ablation essentially produces a smaller-sparse model if I understand correctly, which would decrease the model capacity. Can the authors elaborate more about this? Does SRigL w/o ablation means only pruning these dean channels without weight regrowing?
>
> Your understanding is partially correct, as we ablate neurons the model becomes less wide (fewer neurons / channels). However, since we redistribute the parameters from ablated neurons to active neurons, the target sparsity and active parameters per layer remains fixed throughout the training process. So the model’s capacity itself is arguably fixed, but the smaller-sparse model has more connections per neuron than the wide-sparse model at initialization.
>
> We observed that unstructured DST methods reallocate parameters to specific neurons, without any explicit mechanism for doing so. We speculate that the increased performance from neuron ablation is due to the DST algorithm concentrating its parameter budget on highly relevant and salient features that are present within the active neurons; however, more work is required to confirm this speculation and introduce a more rigorous theoretical basis for understanding this phenomena. We hope to follow this work up with such an analysis in the future.

---

> ### Author Response · Authors · 2023-11-17
> **Rebuttal to 43nv (2/2)**
>
> >(1) What is the technical difference between SRigL and Chase?
>
> See above.
>
> > (2) Can SRigL also accelerate the training process for the online inference, with real-world wall-clock saving? Moreover, I noticed that the Constant Fan-in sparsity can be accelerated by GPUs with some custom CUDA implementation, such as Schultheis & Babbar (2023). I am wondering how difficult to make SRigL accelerated by GPUs using the CUDA implementation provided by Schultheis & Babbar (2023).
>
> Yes, the hybrid structured / fine-grained sparsity learned by SRigL can be accelerated during training, unlike purely unstructured DST methods or typical N:M methods. Training with fine-grained structure requires transposable constraints as per [1]. During inference, we can exploit Schultheis & Babbar’s (2023) GPU kernels with ease. *We are adding GPU benchmarking using these kernels to our paper and these will be available before the close of the discussion period.*
>
> > (3) Why SRigL w/ ablation outperforms SRigL w/o ablation? I suppose that SRigL w/o ablation is essentially the unstructured version of RigL.
>
> See above for our speculation on the cause. However, we note that SoTA unstructured DST methods *already ablate neurons*; therefore, we find empirically that enabling such ablation in more structured settings such as SRigL benefits overall accuracy and runtime characteristics of the sparse network.
>
> One point of clarification, SRigL without ablation still has the constant fan-in constraint applied to it which is a significant difference compared to RigL, which learns a purely unstructured sparse mask.
>
> > (4) Since the real-world clock-time is measured on CPU. I am wondering how different it is to implement GPU kernel to support SRigL in common GPUs?
>
> Thanks to Schultheis & Babbar’s (2023), we can report these timings now. See above. These kernels work on existing commodity hardware without any specialized requirements.
>
> [1] I. Hubara, B. Chmiel, M. Island, R. Banner, J. Naor, and D. Soudry, “Accelerated Sparse Neural Training: A Provable and Efficient Method to Find N:M Transposable Masks,” in Advances in Neural Information Processing Systems, Curran Associates, Inc., 2021, pp. 21099–21111.

---

> > ### Comment · Reviewer_43nv · 2023-11-20
> > **My concerns are all addressed**
> >
> > Dear Authors,
> >
> > I thank the authors for the detailed and honest answers to my question. I love the fact you acknowledge what is lacking to accelerate training with SRigL. I would like to see the discussion with Chase in the camera ready version.
> >
> > I believe this work provides a good example for hardware-friendly sparse training. I look forward to seeing the follow-up work on training speedup.
> >
> > Overall, I am happy with the rebuttal. I will increase my score to 8 and push for an acceptance.

---

### Author Response · Authors · 2023-11-17
**General Rebuttal & Updated Manuscript**

We thank our reviewers for their constructive criticism and helpful comments. A revised copy of our paper has now been uploaded which includes the following additions requested by the reviewers:


* New MobileNet experiments on Imagenet. We report initial results for SRigL and RigL with MobileNet-V3 small and large. These runs remain in progress as the training procedure calls for 600 epochs. We will continue to update the paper as training progresses.
* In-time over parameterization rates for RigL and SRigL on a variety of datasets and models
* GPU benchmark timings using kernel implementation by Schultheis & Babbar (2023)
* Added a comparison to NeuralMagic’s DeepSparse engine for unstructured sparsity
*  Added AST results to Table 3
* Added comparison with structured pruning methods in Table 10 at the end of the appendices.

The above-noted additions (except for AST results) have been included after Appendix H to differentiate them from our originally submitted work. We intend to add further details regarding these new results for the camera-ready version.

We have endeavored to respond to the various weaknesses and questions posed by our reviewers below. However, we’d like to address selected comments that were received y more than one reviewer:

* *Lack of comparison with Yin, Lu, et al. "Dynamic Sparsity Is Channel-Level Sparsity Learner"*: This work does include some similar observations that we make regarding neuron ablation / pruning amenable channels. However, as noted below there are significant differences between our work and Lu et al.’s work. Further, while we will reference this work in our camera-ready submission, it’s worth noting that it is a contemporaneous work as it has only been published within the last 2 months.
* *Concern regarding lack of novelty*: We believe that we offer several novel contributions included in this work that are worth publishing and sharing with the broader sparse neural network community:
  - We are the first to demonstrate that the specific fine-grained sparsity produced by a constant fan-in constraint yields sparse models with generalization performance that is comparable to unstructured sparsity. While constant fan-in is a subset of N:M sparsity, there are some key differentiators as outlined in our paper and rebuttals below.
  - Our neuron ablation methodology and empirical results highlight key differences between current SOTA unstructured and structured / fine-grained sparse networks. We believe that our observations relating to the ablation of neurons inherent with RigL, i.e. the fact that RigL learns to reduce layer width at high sparsity, is of significant interest to the DST research community, and will lead to future work that further explores the relationship between sparsity and layer width in DST methods in general.
  - We are the first to demonstrate that *sparse-to-sparse* Dynamic Sparse Training (DST) algorithms can be modified to find a performant variant of N:M fine-grained structured sparse networks, i.e. without affecting the state-of-the-art generalization performance of unstructured DST methods.
  - We demonstrate real-world acceleration using a sparse layer obtained from SRigL on CPU for online inference and GPU for batched inference. These results compare favorably to other acceleration strategies such as the DeepSparse Engine, CSR, and purely structured sparsity.

We think it fair to say that our paper's method, empirical evaluation, and motivation themselves were well received in general by all the reviewers. We believe the paper provides actionable insight and new research questions that will be of interest to the research community in both sparse training methods and hardware acceleration of sparse NN representations.

We encourage the reviewers to ask further questions in the discussion phase that will aid in their decisions, and are also happy to further address any outstanding concerns. We are still working on some of the requested additional results and experiments, and will update the paper as additional results are made available.

---

### Meta-Review · Area_Chair_WBJm · 2023-12-06

**Metareview:**

The paper proposes SRigL, a method for dynamic sparse training of neural networks. This is an extension of RigL, which is among the better performing methods for sparse neural network training (to date) -- the main innovation here is the imposition of structured (N:M) sparsity in the neuron weight patterns (as opposed to RigL which used standard sparsity constraints). N:M sparsity is known to to be suitable for hardware acceleration, so the method can lead to significant (potential) future impact.

The paper initially received mixed reviews. Reviewers raised issues surrounding a lack of demonstrations on low-redundancy networks (such as MobileNet), lack of comparisons with existing methods such as Chase, and insufficient evidence that the method works significantly better than RigL. There was a fairly productive rebuttal period in which the authors were able to address many of the above questions.

Issues surrounding the evaluations still remain. In particular, systematic evaluations showing end-to-end training speedups (in practice on GPUs, not just in theory) of SRigL over RigL are still absent from this paper. Beefing up this section significantly might be a point in favor of the paper in terms of potential future impact. The paper could benefit from a last round of revision addressing this aspect.

**Justification For Why Not Higher Score:**

Paper is on the borderline, with somewhat limited methodological contributions.

**Justification For Why Not Lower Score:**

The paper does make a non-trivial advance in the (specific) line of literature centered around dynamic sparse training, and as such is well written.

---

### Decision · Program_Chairs · 2024-01-16

Accept (poster)